# Neurocognitive processing efficiency for discriminating human non-alarm rather than alarm scream calls

**Sascha Frühholz** [1,2,3,4]*, **Joris Dietziker**[1], **Matthias Staib**[1], **Wiebke Trost**[1]

**1** Cognitive and Affective Neuroscience Unit, University of Zurich, Zurich, Switzerland, **2** Neuroscience Center Zurich, University of Zurich and ETH Zurich, Zurich, Switzerland, **3** Department of Psychology, University of Oslo, Oslo, Norway, **4** Center for the Interdisciplinary Study of Language Evolution, University of Zurich, Zurich, Switzerland

* sascha.fruehholz@uzh.ch

**Data Availability Statement:** All relevant data from the behavioral experiments are within the paper and its Supporting Information files. Main fMRI contrasts can be found at https://identifiers.org/neurovault.collection:9449.

## Abstract

Across many species, scream calls signal the affective significance of events to other agents. Scream calls were often thought to be of generic alarming and fearful nature, to signal potential threats, with instantaneous, involuntary, and accurate recognition by perceivers. However, scream calls are more diverse in their affective signaling nature than being limited to fearfully alarming a threat, and thus the broader sociobiological relevance of various scream types is unclear. Here we used 4 different psychoacoustic, perceptual decision-making, and neuroimaging experiments in humans to demonstrate the existence of at least 6 psychoacoustically distinctive types of scream calls of both alarming and non-alarming nature, rather than there being only screams caused by fear or aggression. Second, based on perceptual and processing sensitivity measures for decision-making during scream recognition, we found that alarm screams (with some exceptions) were overall discriminated the worst, were responded to the slowest, and were associated with a lower perceptual sensitivity for their recognition compared with non-alarm screams. Third, the neural processing of alarm compared with non-alarm screams during an implicit processing task elicited only minimal neural signal and connectivity in perceivers, contrary to the frequent assumption of a threat processing bias of the primate neural system. These findings show that scream calls are more diverse in their signaling and communicative nature in humans than previously assumed, and, in contrast to a commonly observed threat processing bias in perceptual discriminations and neural processes, we found that especially non-alarm screams, and positive screams in particular, seem to have higher efficiency in speeded discriminations and the implicit neural processing of various scream types in humans.

## Introduction

Vocal affect bursts, such as crying, grunting, or laughing, are a major part of sociobiological communication across many mammalian species, especially primates. A specific type of affect

**Funding:** This study was supported by the Swiss National Science Foundation (SNSF PP00P1_157409/1 and PP00P1_183711/1 to SF). The funders had no role in study design, data collection and analysis, decision to publish, or preparation of the manuscript.

**Competing interests:** The authors have declared that no competing interests exist.

**Abbreviations:** 1w-ANOVA, 1-way analysis of variance; 2AFC, 2-alternative forced-choice; 7AFC, 7-alternative forced-choice; BMA, Bayesian model averaging; DCM, dynamic causal modeling; FDR, false discovery rate; fMRI, functional magnetic resonance imaging; GLM, general linear model; IFC, inferior frontal cortex; ITI, inter-trial interval; MNI, Montreal Neurological Institute; MPS, modulation power spectrum; mSTG, middle superior temporal gyrus; mSTS, middle superior temporal sulcus; PPo, planum polare; pSTS, posterior superior temporal sulcus; ROI, region of interest; RT, response time; SPL, sound pressure level; SVM, support vector machine.

burst is a scream call. Screams are relatively short, loud and intense, high-pitched, tremulous, and rough voice calls [1–3]. They have a far-reaching impact [1,4] and seem to be immediately recognized, adaptively and rapidly responded to, and hard to ignore by perceivers [5,6]. Screams are thus assumed to be an effective mode of communicating affect signals that are of high relevance in any sociobiological interaction [1]. Scream calls have a long evolutionary trajectory across many species up to human primates. In nonhuman primates [7–9] and other mammalian species [10], scream-like calls are frequently a specific type of alarm signal used exclusively in negative contexts, such as in in-group social conflicts between animals of different rank. Screams by lower-ranking animals help to recruit support from allies [11,12], while higher-ranking animals scream to recruit support from matrilineal kin when challenged by lower-ranking opponents [13]. Furthermore, "SOS screams" signal the presence of environmental threats (e.g., predators) [14,15]. Scream-like voice calls thus aim to trigger certain behavior in potential listeners, similar to other types of alarm calls that are commonly expressed in high-arousal states of fear [16] or aggressiveness [17].

Accordingly, human screams are assumed to share this acoustical and motivational nature with the screams of other species, highlighting the alarm quality of such affect bursts to signal danger and to scare potential predators [1,4]. Being of alarming nature, screams thus demand urgent responses in listeners, implying a fast recognition and an accurate perceptual categorization [5,18] as well as neural efficiency in processing [1]. Previous research on alarm signal perception indicated that alarm signals induce strong physiological and alertness reactions in listeners [19,20] that facilitate the detection of other visual and auditory targets [21,22], especially with increasing levels of urgency perceived in the primary alarm signals [21,22]. However, research is lacking that quantifies the response time (RT) and the accuracy of processing the alarm signal itself, as well as the effects on the detection of secondary signals. Investigating response speed and accuracy in the perception of screams as a unique type of alarm signal might be an ideal way to obtain data on the processing efficiency of alarm signals themselves. This fast and efficient processing of the alarm quality of screams might be based on a certain acoustic scream feature, since a previous report [1] suggested that a specific frequency range of the temporal modulation rate was related to the perceptual feature of "roughness" (i.e., a harsh, thrilling, croaky sound) in the broad 30- to 150-Hz range that contributes to the alarm quality of fear screams [23]. Other studies confirmed that roughness, amongst other important acoustic and perpetual features, is a defining perceptual feature of screams that enables listeners to classify vocalizations as screams [3].

Although previous studies thus provided a detailed functional and neural description of scream call recognition, they were limited from 3 perspectives. The first limitation concerns the variety of primate screams, especially human screams. These studies largely focused on alarming fear screams [1,5,10] and sometimes on scream-like and alarming aggressive roars [17]. However, humans scream not only when they are fearful and aggressive, but also when they experience other affective states, and these diverse states that elicit screams are similar to a variety of inner emotional states that are more commonly expressed in less intense nonverbal vocal emotions [24], such as sadness, joy, and sensual pleasure. The second limitation is that, although some previous studies seem to have investigated a broad variety of scream-like vocalizations, these vocalizations seemed to be intermingled with other non-scream-like vocalizations [25] or partly collected from mixed sources [3,25]. The third limitation concerns the neural mechanisms and dynamics of scream perception, as only a limited description of neural mechanisms for scream perception is as yet available [1]. To overcome these limitations, we here aimed to provide a broader description of human screams on 3 interdependent levels that are relevant for any kind of vocal communication. These 3 levels are represented by (a) the acoustic properties and acoustic distinctiveness of scream calls as produced by a sender

(experiment 1), (b) the perceptual and categorization dynamics of these scream calls in listeners (experiments 2 and 3), and (c) the neural mechanisms of scream recognition in the central nervous system of listeners (experiment 4).

Besides being motivated by these aforementioned limitations, the experiments described here were also motivated by the assumption that a broader taxonomy of distinct human screams potentially needs to distinguish at least (a) screams when experiencing intense pleasure, (b) desperate scream-like cries during sadness, (c) screams of joy and elation, (d) screams of pain, (e) screams of anger and rage, and (f) fearful screams (Fig 1; S1 Fig). This broader taxonomy of screams is proposed based on a survey of many daily natural, social, and cultural manifestations of human screams, and the general diversity of human affective vocalizations [26] and especially of nonverbal expression of emotions [24]. While we assume that screams d–f are of an alarming nature (i.e., they call for immediate action), screams a–c are largely non-alarming (i.e., no fast response required; see also below). Furthermore, while screams of joy and pleasure are of a positive nature, the other 4 types of screams have a negative affective valence.

## Results

### Psychoacoustic and affective diversity of human screams

Given the assumed communicative relevance of diverse types of scream calls, they should have a distinct and differential acoustic profile during their expression in senders as the basis for any consequent perception in listeners. Concerning this acoustic and expressive nature of screams, we accordingly performed a first experiment (experiment 1) wherein we asked

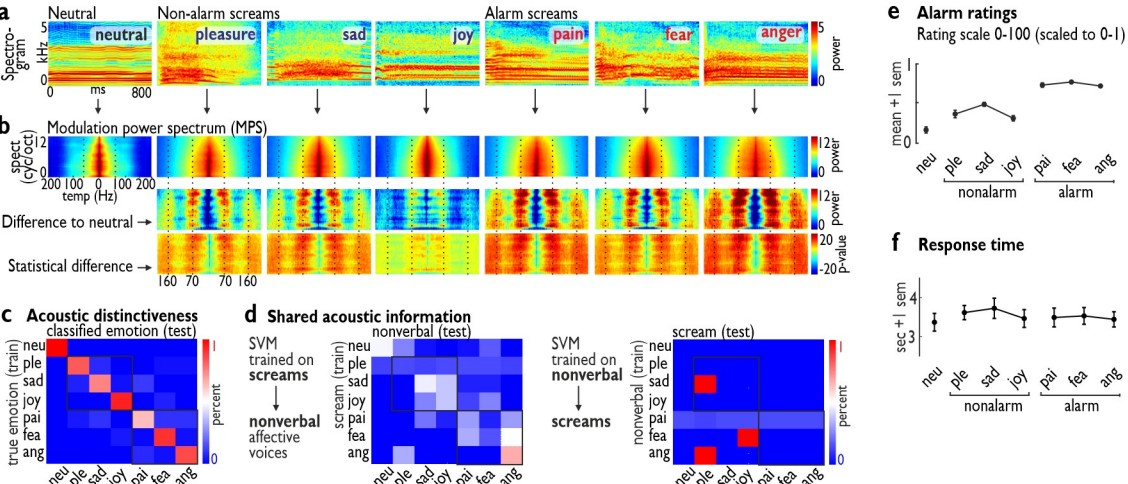

**Fig 1. Acoustic description and alarm ratings of 6 distinctive types of screams.** (a) Spectrograms of example stimuli for each scream type, including a neutral scream of an intense vocalization of the vowel /a/. (b) Average modulation power spectrum (MPS) for each scream type (upper row) as the numeric (middle row) and statistical difference (lower row; n = 60 per scream type; n = 420 screams in total) of each scream type from neutral screams. (c) Using a support vector machine (SVM) on the acoustic features of each scream allowed us to separate each scream type from other screams with high accuracy (e.g., low level of off-diagonal misclassifications). (d) To test whether screams share acoustic information with common nonverbal affect bursts, we trained the SVM either on the screams (left) or on the nonverbal affect bursts (right) and tested the classifier on the other type of affective vocalizations; in both cases, the SVM classifier failed to show superior performance (e.g., high level of off-diagonal misclassifications). (e) Alarm level ratings (n = 23) confirmed the differential level of alarm of screams that seem to be of a low alarming nature (non-alarm screams: pleasure, sadness, joy) and a high alarming nature (alarm screams: pain, fear, anger). (f) Response times for the alarm ratings as performed in a non-speeded task by moving a slider on a visual analog scale to the appropriate alarm level for each scream. Numerical data underlying the plots (c–f) can be found in S1 Data. ang, anger; fea, fear; neu, neutral; pai, pain; ple, pleasure; sad, sadness; sem, standard error of the mean; test, testing data; train, training data.

human participants to vocalize a broad variety of positive and negative screams from instructions based on a collection of typical situations that would elicit each type of scream (Fig 1A–1D). Participants ($n$ = 12) vocalized these 6 types of screams in addition to a neutral but intense vocalization of the vowel /a/. This last category of neutral vocalizations (referred to as "neutral screams" for the sake of terminological simplicity) was included as a baseline and comparison condition to quantify acoustic, perceptual, and neural differences in processing the 6 vocal scream types. The recordings of the neutral and the 6 scream types resulted in a total of 420 acted scream calls that were acoustically similar to natural screams [1,2]. The screaming quality of these vocalizations was verified by ear by trained senior members of the research team, since clear formal acoustic criteria for scream calls are not yet established [3]. Scream calls are also easily identified and discriminated from other nonverbal affective vocalizations [25].

To test whether all screams are always of a highly alarming nature, we asked an independent sample of participants ($n$ = 23) to rate the alarm quality of each scream (Fig 1E and 1F). Although the term "alarm" often refers to certain sociobiological contexts and events that elicit screams, especially in animal settings, the scream can be regarded as the sender's expression of the alarm significance of the context or event. Listeners then rate the alarm level of these screams as part of their decision of how urgently one needs to respond. We quantified these listeners' alarm ratings individually for each of the 420 screams and on a dimensional basis (i.e., independent of the knowledge to which category the scream belonged). As hypothesized, screams differed significantly in their alarm quality (1-way analysis of variance [1w-ANOVA], 7 levels; $F_{6,132}$ = 113.95, $p < 0.001$, $\eta^2$ = 0.77; Greenhouse–Geisser correction applied to all $p$-values in case of sphericity violations based on a Mauchly's test), such that alarm screams (pain, anger, fear) were overall significantly more alarming (1w-ANOVA, 3 levels; $F_{2,44}$ = 227.63, $p < 0.001$, $\eta^2$ = 0.84) than non-alarm screams (pleasure, sadness, joy) ($p < 0.001$) and neutral screams ($p < 0.001$) (Fig 1E). We therefore categorized the pain, anger, and fear screams as "alarm screams" and pleasure, sadness, and joy screams as "non-alarm screams." This categorization was based on a full permutation and statistical testing through any possible categorical combination of the 6 distinctive scream types. Categorizing all screams into neutral screams, non-alarm screams (pleasure, sadness, joy), and alarm screams (pain, anger, fear) led to the overall most significant result (1w-ANOVA, 3 levels; $F_{2,44}$ = 227.63, $p < 0.001$, $\eta^2$ = 0.84), while any other kind of combination was less significant ($F_{2,44}$ values 114.24–225.91). The chosen categorization thus maximized the difference between the scream categories along the alarming dimension the best, followed by the second-best categorization of only fear and anger as alarm screams ($F_{2,44}$ = 225.91), and the third-best categorization of only fear and pain as alarm screams ($F_{2,44}$ = 221.23).

In an additional analysis, we tested for RT difference for the alarm ratings (Fig 1F) and found no difference across all 7 types of screams ($F_{6,132}$ = 2.192, $p$ = 0.098, $\eta^2$ = 0.01) nor between the 3 major categories (neutral, alarm, and non-alarm) ($F_{2,44}$ = 3.015, $p$ = 0.084, $\eta^2$ = 0.01). Thus, these self-paced alarm ratings were performed with a similar response latency across scream types, but with overall relatively slow responses (>3 s) compared to the speeded classification tasks we report below.

Since all screams were normalized to a uniform loudness sound pressure level (SPL) of 70 dB, we furthermore tested whether this loudness normalization influenced the alarm ratings of the screams. For each scream, we quantified the loudness level before and after normalization, calculated the loudness difference between the original and the normalized loudness, and performed a correlation analysis with the alarm ratings for each scream, which yielded a non-significant relationship (Pearson correlation, $r$ = −0.071, $p$ = 0.146). This indicates that the alarm ratings were not influenced by the loudness normalization procedure.

Besides this perceptual alarm rating of the scream calls, we also performed an acoustic analysis of each type of scream. In a first acoustic analysis, we calculated the spectrogram of each of the alarm and non-alarm screams (Fig 1A) and then quantified the modulation power spectrum (MPS) [27,28] of the 3 alarm and the 3 non-alarm screams (Fig 1B). The MPS is an estimate of the spectral and temporal information in the acoustic scream signal and was calculated similarly to a previous study [1]. We found that these temporal modulation rates were focused around frequency bands centered on a low level of approximately 60 Hz (range 50–70 Hz) and a high level of approximately 160 Hz (range 140–180 Hz), and this was characteristic both of alarm screams (fear, pain, anger) and, most importantly, also of some non-alarm screams (pleasure, sadness). All scream types seem to have a broader frequency range for temporal modulations, and the specific comparison to neutral screams especially highlighted these 2 frequency ranges. These modulation rates for the temporal evolvement of sound have been thought to be associated with the perceptual features of "roughness" [1]. Although this roughness is not an exclusive feature of screams, as many other nonverbal vocalizations can have a rough acoustic attribute [25], it is nonetheless very characteristic of screams. Specifically, the power of the modulation rate quantified for each scream in the low- (ß = 15.665) and high-frequency range (ß = 4.903) predicted positively the alarm ratings for each scream as quantified by a linear regression analysis ($F = 138.240$, $p < 0.001$, $R^2 = 0.399$).

Many types of alarm and non-alarm screams thus seem to share a rough acoustic property, but roughness is only 1 feature among many other important voice features. We therefore asked in a second step of the acoustic analysis whether scream types are acoustically different overall beyond this roughness feature. We analyzed each scream type according to a larger and well-established set of 88 acoustic features [29], including a proper tracking of acoustic pitch. These 88 features are centrally related to a general acoustic voice space. We subjected these features to a machine learning approach implemented as a support vector machine (SVM). The SVM was trained (training data) on the acoustic features of a subset of the screams, and then was tested (testing data) to classify a new set of screams into 7 possible categories (Fig 1C). The overall classification accuracy in this cross-validation approach was high, at 79.8% (chance level 14.3%), with accuracies ranging from 65.8% (sadness) to 89.7% (joy) and 100% (neutral). This confirms that the scream types are acoustically very distinctive from each other at a high level, and also that non-alarm screams are as distinctive as alarm screams, within and across these larger scream categories.

We finally also tested whether the 6 distinctive scream types are acoustically distinct from more basic nonverbal affect vocalizations of similar affective valence, such as laughs, cries, or anger growls [30]. In this cross-classification approach, the SVM trained on scream calls was unable to classify nonverbal vocal affect (overall classification accuracy 14.3%, chance level 14.29%), and the acoustic features of nonverbal affective voices did not help with classifying screams (overall accuracy 19.2%) (Fig 1D).

## Low perceptual sensitivity for alarm screams during speeded classifications

The expression of scream calls by senders is the basis for their communication and for the potential detection and recognition by receivers. Having shown in experiment 1 that human screams are not limited to alarm screams signaling threat, but rather show an acoustic diversity of at least 6 different types of screams, we investigated in experiment 2 how accurately human listeners could perceptually discriminate and classify these different scream types in a speeded classification task. We presented a selection of 84 of the original 420 scream calls. All selected screams had an equal base recognition rate to ensure that different accuracy rates for different scream types in this experiment were not biased by a different base recognition rate of the

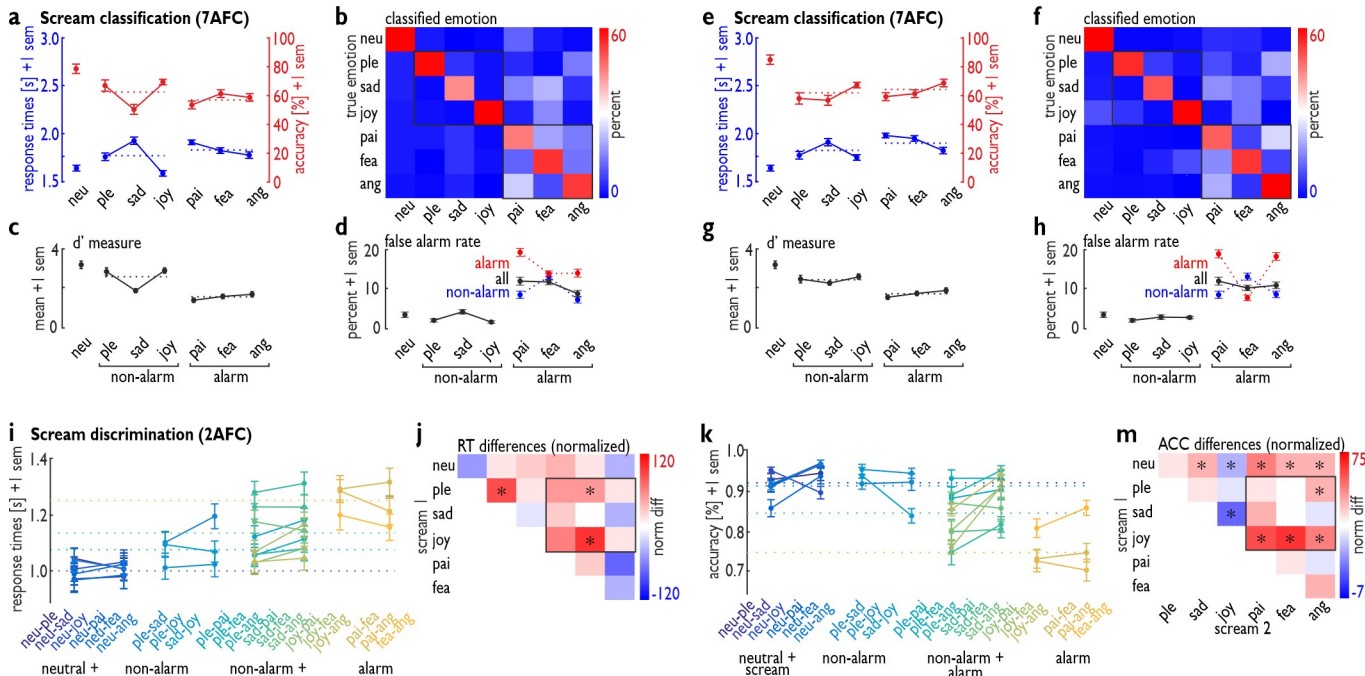

**Fig 2. Perceptual decision-making on perceived scream calls.** (a–d) RTs and accuracy level for (a) categorizing the 7 types of screams in a 7AFC task (top left) in experiment 2 (*n* = 33). (b) During misclassification of screams (top right, off-diagonal), participants significantly used more categories from alarm screams than from non-alarm screams (bottom right). (c) Combining the hit rate and the false alarm rate results in the *d'* measure of sensitivity to detect a certain scream type (bottom left). (d) The false alarm rate is also reported separately for alarm (red) and non-alarm screams (blue). (e–h) The experiment 2 was replicated with an independent sample of *n* = 29 participants but using a new selection of 84 scream stimuli. The data were analyzed identically to the data reported in (a–d). (i–m) RT (i) and accuracy level (k) for discriminating scream calls (experiment 3, *n* = 35) from neutral screams (first set), for discriminating non-alarm screams from non-alarm screams (second set), for discriminating alarm screams from non-alarm screams (third set), and for discriminating alarm screams from alarm screams (fourth set). Panels (j) and (m) show the normalized RT and accuracy difference between the 21 combinations of screams. *p* < 0.05, based on a non-parametric permutation test. Numerical data underlying the plots (a–m) can be found in S2 Data. 2AFC, 2-alternative forced-choice; 7AFC, 7-alternative forced-choice; ang, anger; fea, fear; neu, neutral; norm diff, normalized difference; pai, pain; ple, pleasure; sad, sadness; sem, standard error of the mean.

selected stimuli. These selected screams were presented to another sample of human listeners (*n* = 33), who were asked to classify the screams into 7 categories that referred to the 6 scream types and neutral vocalizations (Fig 2A–2H). Since also this selection of 84 screams were normalized to a loudness level of 70 dB SPL, we tested if this normalization procedure affected the scream types differently. The loudness difference score between original and normalized screams did not differ across scream types (1w-ANOVA, 7 levels; $F_{6,66} = 1.375$, $p = 0.238$, $\eta^2 = 0.09$), and thus was unlikely to affect the behavioral data reported below.

RTs for correctly classified trials (1w-ANOVA, 7 levels; $F_{6,192} = 26.049$, $p < 0.001$, $\eta^2 = 0.29$) and performance accuracy rates (1w-ANOVA, 7 levels; $F_{6,192} = 12.273$, $p < 0.001$, $\eta^2 = 0.212$) significantly differed across the 7 scream types (Fig 2A). Post hoc tests on RTs showed that all scream types were classified slower than neutral screams (all $p < 0.035$), except for joy screams ($p = 0.487$); all non-alarm screams had different RTs (all $p < 0.006$), while only pain and anger differed ($p = 0.005$) in RTs for alarm screams. Comparisons between non-alarm and alarm screams showed that joy screams were classified faster than all alarm screams (all $p < 0.001$), and pleasure screams were classified faster than pain screams ($p = 0.013$). Only sadness screams were classified slower than anger screams ($p = 0.013$). Post hoc tests for the performance accuracy showed that neutral screams had a better classification rate than the negative screams (all $p < 0.001$), but not compared to the positive screams (all $p > 0.091$); within the non-alarm screams, sadness screams differed from pleasure ($p = 0.034$) and joy screams ($p <$

0.001), while alarm screams did not differ in performance accuracy (all $p > 0.142$). Comparisons between non-alarm and alarm screams showed that joy screams were more accurately classified than anger ($p = 0.004$) and pain screams ($p < 0.001$).

To better characterize these differences on a broader level, we repeated the analysis with only 3 categories on the basis of the aforementioned observation that scream calls largely differ in their alarm level. The scream calls were divided into 3 major categories, represented by the neutral screams, non-alarm screams (pleasure, sadness, joy), and alarm screams (pain, fear, anger), and we calculated the mean RTs and classification accuracies for each participant within these 3 categories. These 3 categories differed in RTs (1w-ANOVA, 3 levels; $F_{2,64} = 27.680$, $p < 0.001$, $\eta^2 = 0.20$), with non-alarm ($p < 0.001$) and alarm screams ($p < 0.001$) having slower mean RTs than neutral screams, and non-alarm screams being classified faster than alarm screams ($p = 0.005$). The latter difference however seems strongly driven by a fast response to joy screams, and partly also to pleasure screams, compared to the 3 alarm screams. The 3 categories also differed in accuracy rates (1w-ANOVA, 3 levels; $F_{2,64} = 28.034$, $p < 0.001$, $\eta^2 = 0.30$), with lower accuracy for non-alarm ($p < 0.001$) and alarm screams ($p < 0.001$) than for neutral screams, but with no difference in accuracy between non-alarm and alarm screams ($p = 0.183$) (Fig 2A).

In addition, we quantified the false alarm rate, meaning how often each scream category was selected during misclassifications (Fig 2B–2D). This analysis figures as an indicator of the decisional relevance of scream categories in subjective classifications and misclassifications. The frequency of selecting alarm scream categories during misclassifications was higher (1w-ANOVA, 3 levels; $F_{2,64} = 78.602$, $p < 0.001$, $\eta^2 = 0.44$) compared with that for the neutral ($p < 0.001$) and non-alarm categories ($p < 0.001$) in the comparison of 3 major scream categories (neutral, non-alarm, alarm) (Fig 2B). Thus, participants tended to perceive screams as "alarming" when they misclassified scream calls, that is, they more likely chose 1 of the alarming scream types during misclassification. This was true during misclassifications of both non-alarm screams (i.e., classified in 1 of the 3 alarm scream types) and especially of alarm screams (i.e., classified in 1 of the other 2 alarm scream types instead of the target alarm scream type) (Fig 2D).

By considering the false alarm rate for using a certain scream type for decisional misclassification in relation to the correctly identified scream for this category, we calculated the $d'$ measure as an indicator of perceptual sensitivity to a certain scream type for the 6 scream types (Fig 2C). All scream types differed in their perceptual sensitivity across the 7 categories (1w-ANOVA, 7 levels; $F_{6,192} = 62.527$, $p < 0.001$, $\eta^2 = 0.53$), with alarm screams revealing the lowest perceptual sensitivity scores when comparing the 3 major categories (1w-ANOVA, 3 levels; $F_{2,64} = 74.021$, $p < 0.001$, $\eta^2 = 0.37$). We have to note that this pattern of results is unlikely to be driven by the acoustic similarity or dissimilarity of some of the 6 scream types, as the machine learning approach discussed above has shown a relatively large acoustic distance (i.e., high above chance discrimination by the machine classifier; Fig 1D) between all types of screams.

To test the replicability of the patterns of results described above in the speeded classification of a selection of 84 screams of the total of 420 screams, we replicated the same study again, but using a different selection of 84 screams that minimally overlapped with the first selection of 84 screams (Fig 2E–2H). We asked another and independent sample of $n = 29$ participants to perform the identical task on this second selection of 84 screams. We found an almost identical pattern of results as found for the first selection of screams. RTs in correctly classified trials significantly differed across the 7 scream types (1w-ANOVA, 7 levels; $F_{6,168} = 17.606$, $p < 0.001$, $\eta^2 = 0.29$). Post hoc tests on RTs showed that all scream types were classified slower than neutral screams (all $p < 0.029$), except for joy screams ($p = 0.186$); among non-alarm

screams, RTs for sadness screams were different from those for joy ($p < 0.001$) and pleasure screams ($p = 0.018$); among alarm screams, RTs for anger screams were different from those for fear ($p = 0.014$) and pain screams ($p = 0.007$). Comparisons between non-alarm and alarm screams showed that joy screams were classified faster than pain ($p < 0.001$) and fear screams ($p = 0.001$), and pleasure screams were classified faster than pain ($p = 0.002$) and fear screams ($p = 0.005$). Furthermore, the 3 categories differed in RTs (1w-ANOVA, 3 levels; $F_{2,56} = 30.075$, $p < 0.001$, $\eta^2 = 0.36$), with non-alarm ($p < 0.001$) and alarm screams ($p < 0.001$) having slower mean RTs than neutral screams, and non-alarm screams being classified faster than alarm screams ($p = 0.004$).

Performance accuracy rates significantly differed across the 7 scream types (1w-ANOVA, 7 levels; $F_{6,168} = 11.326$, $p < 0.001$, $\eta^2 = 0.23$). Post hoc comparisons showed that all scream types were classified worse than neutral screams (all $p < 0.001$), with only sadness being classified worse than anger screams ($p = 0.044$). The 3 major categories also differed in accuracy rates (1w-ANOVA, 3 levels; $F_{2,56} = 43.293$, $p < 0.001$, $\eta^2 = 0.44$), with lower accuracy for non-alarm ($p < 0.001$) and alarm screams ($p < 0.001$) than for neutral screams, but with no difference in accuracy for non-alarm and alarm screams ($p = 0.755$) (Fig 2A). The false alarm rate was different across all scream categories (1w-ANOVA, 7 levels; $F_{6,168} = 30.241$, $p < 0.001$, $\eta^2 = 0.48$), and the frequency of selecting alarm scream categories during misclassifications was higher (1w-ANOVA, 3 levels; $F_{2,56} = 35.133$, $p < 0.001$, $\eta^2 = 0.38$) compared with that for the neutral ($p < 0.001$) and non-alarm categories ($p < 0.001$) in the comparison of the 3 major scream categories. Finally, all scream types differed in their perceptual sensitivity (1w-ANOVA, 7 levels; $F_{6,168} = 40.673$, $p < 0.001$, $\eta^2 = 0.47$), with alarm screams revealing the lowest perceptual sensitivity scores when comparing the 3 major categories (1w-ANOVA, 3 levels; $F_{3,56} = 93.452$, $p < 0.001$, $\eta^2 = 0.49$), and when alarm screams were compared to non-alarm ($p < 0.001$) and neutral screams ($p < 0.001$). These data from the replication of experiment 2 thus indicate that the pattern of results seems largely similar across variable selections of subsamples of 84 screams out of the 420 total screams.

## Impaired perceptual discrimination of alarm compared to non-alarm screams

In experiment 2, the 6 scream calls as well as the 3 major categories of scream types (neutral, non-alarm, alarm) showed some selective differences in behavioral and sensitivity measures of their classification when all types of screams and choice options were available in a speeded multi-option decision task. Alarm screams showed lower neurocognitive processing efficiency (i.e., RT measure) and perceptual sensitivity (i.e., $d'$ measure) than non-alarm screams. This surprising pattern of a relatively higher processing efficiency for non-alarm than alarm screams might be specific to the more complex 7-alternative forced-choice (7AFC) task of experiment 2. This pattern might be different in simpler speeded discrimination tasks while classifying only a low number of possible scream types. Specifically, in daily life, individuals often need to choose between only 2 scream type options depending on certain contexts. In threatening situations, for example, individuals need to discriminate fear from anger, but not from other types of screams.

In experiment 3, therefore, we had another sample of participants ($n = 35$) discriminate screams in any combination of only 2 types of screams presented in the same block (e.g., joyful and fearful screams), with 21 possible combinations, using a simple 2-option decision task (e.g., classify screams as joyful or fearful) (Fig 2I–2M). We found that only some combinations of screams led to differences in RTs and accuracy between the 2 types of screams presented. Participants responded to pleasure screams ($t$ tests, all false discovery rate [FDR] corrected;

$t_{34} = 3.022$, $p = 0.033$) and joy screams ($t_{34} = 4.858$, $p < 0.001$) faster when these screams were presented with fear screams in 1 block, as well as to pleasure screams when presented with sadness screams ($t_{34} = 3.730$, $p = 0.007$) in 1 block (Fig 2I and 2J). In addition, more discrimination errors occurred for all negative screams (i.e., screams with a negative affective valence) when presented together with neutral screams (sadness: $t_{34} = 3.358$, $p = 0.005$; pain: $t_{34} = 4.980$, $p < 0.001$; fear: $t_{34} = 3.435$, $p = 0.005$; anger: $t_{34} = 3.303$, $p = 0.005$) or with joyful screams (sadness: $t_{34} = 7.615$, $p < 0.001$; pain: $t_{34} = 6.441$, $p < 0.001$; fear: $t_{34} = 7.213$, $p < 0.001$; anger: $t_{34} = 7.973$, $p < 0.001$), for angry screams when presented together with pleasure screams ($t_{34} = 2.652$, $p = 0.025$), and for neutral screams when presented together with joyful screams ($t_{34} = 3.787$, $p = 0.002$) (Fig 2K–2M). This pattern of results suggests that, compared to alarm screams, non-alarm and positive screams again are processed more efficiently in terms of faster RTs and discrimination accuracy in simpler 2-choice decision tasks.

Concerning the above-reported patterns, there might be the possibility that alarm screams do not need to be discriminated very quickly, because they only need to activate the perceiving organisms to indiscriminately respond to any potential threat. While this could explain the increased classification times for alarm screams (i.e., respond first, and then discriminate), it can only partly explain the higher error rates when humans classify and discriminate within alarm screams. An important notion might be that misclassifications with alarm screams often take the form of misclassifying them as another alarm scream, which could lead nonetheless to the same appropriate behavioral adaptions. Furthermore, instead of discriminating within alarm screams, the discrimination between non-alarm and alarm screams might be of higher relevance for survival. We indeed found that discriminating between non-alarm and alarm screams is overall faster and more accurate than discriminating within alarm screams (Fig 2I and 2J). However, although these discriminations between non-alarm and alarm screams showed an overall better performance than discrimination within alarm screams, it was still the alarm screams that revealed worse performance compared to non-alarm screams in specific combinations between non-alarm and alarm screams (Fig 2K–2M). The latter observation still points to some form of classification and discrimination disadvantage for alarm screams.

Given the general distinction of these neutral, non-alarm, and alarm screams, we asked in experiment 3 whether scream discriminations within and across these broader categories would influence participants' performance. We found that discrimination of the 6 scream types from neutral screams led to the overall fastest responses (1w-ANOVA, 4 levels, referring to the 4 possible combinations: neutral versus scream, non-alarm versus non-alarm, alarm versus alarm, and alarm versus non-alarm; $F_{3,102} = 56.449$, $p < 0.001$, $\eta^2 = 0.13$) (Fig 2I) and the highest accuracy rate (1w-ANOVA, 4 levels; $F_{3,102} = 149.090$, $p < 0.001$, $\eta^2 = 0.50$) (Fig 2K) compared with all other combinations. The RT increased and the accuracy rate decreased from discrimination within non-alarm screams to discrimination of non-alarm from alarm screams, and to discrimination within alarm screams. Discrimination among alarm screams required significantly increased RTs (post hoc planned comparisons; all $p < 0.001$) and had significantly decreased accuracy compared with all other combinations (all $p < 0.001$). Again, we have to note that all types of screams were acoustically very distinct from each other given the high above chance level classification of the machine learning approach (Fig 1C), such that discrimination impairments between screams were unlikely driven by potential acoustic similarities.

## Neural efficiency and significance for non-alarm scream processing

With the observation from experiment 2 and 3 that alarm screams are processed with significantly lower neurocognitive efficiency than one would assume given their threat signaling

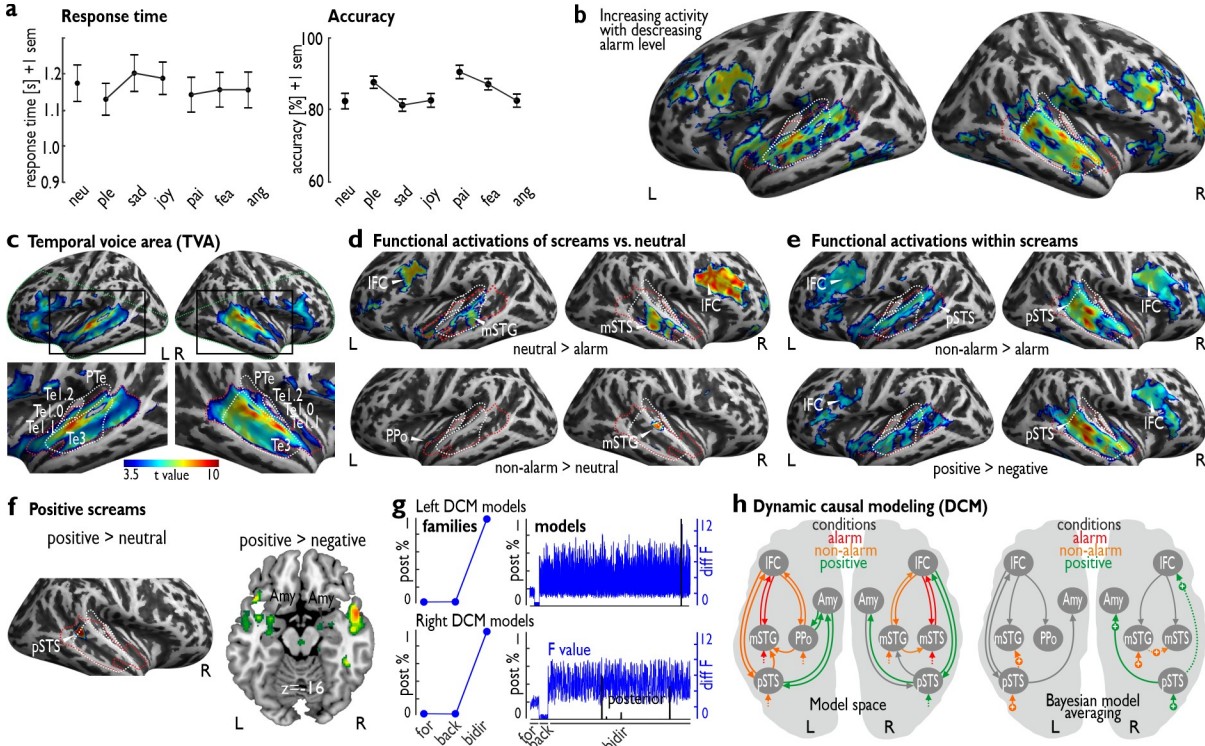

**Fig 3. Neural activity and effective functional network for scream call processing.** (a) Behavioral data from the gender task during the functional magnetic resonance imaging (fMRI) experiment (experiment 4); shown are the response times (left) and the accuracy level (right) for performing a gender decision task on the screams perceived during the fMRI experiment ($n$ = 30). (b) Negative parametric modulation of brain activity by the alarm rating of scream calls; while we did not find neural activity for a positive relationship with the alarm ratings of every scream, we found largely extended activity in the auditory cortex and the frontal cortex for negative associations with the alarm level of each scream. (c) Functional definition of the temporal voice area (TVA) in $n$ = 30 human participants. The red dashed line indicates the cortical extension of the TVA; the green dashed line indicates the coverage of the partial volume acquisition; the white dashed line defines the anatomical subregions of the auditory cortex (primary areas Te1.0, Te1.1, Te1.2, and secondary area Te3) and the planum temporale (PTe). (d) Alarm screams compared with neutral screams ($n$ = 30) elicited reduced activity in the bilateral IFC and STC (left mSTG, right mSTS), whereas non-alarm screams elicited increased activity in the bilateral STC (left PPo, right mSTG). (e) Compared with alarm screams, non-alarm screams elicited significantly higher and extended activity in the bilateral IFC and STC. (f) Positive screams (pleasure, joy) showed higher activity in the right pSTS compared with that for neutral screams and in the bilateral amygdala compared with that for negative screams (sadness, pain, fear, anger). (g) Dynamic causal modeling (DCM) revealed the bidirectional model family as the winning family based on the posterior probability (left panel) in both the left (top left) and right hemispheres (bottom left). The right panel shows the posterior probabilities (black) and the log-evidence (blue; "diff F" = $F$-value minus the minimum $F$-value across models) for each model from the forward (for), backward (back), and bidirectional (bidir) model families. (h) Bayesian model averaging in the bidirectional model family indicated specific significant modulation of the left ($n$ = 28) and right hemisphere ($n$ = 28) only in the non-alarm (orange) and positive (green) scream conditions (see S3 Table). All activations are thresholded at $p$ = 0.005 voxel level and a minimum cluster size of $k$ = 42, resulting in a corrected threshold of $p$ = 0.05 at the cluster level. Numerical data underlying the plots in (a) can be found in S3 Data and S3 Table. Amy, amygdala; ang, anger; fea, fear; IFC, inferior frontal cortex; L, left; mSTG, middle superior temporal gyrus; mSTS, middle superior temporal sulcus; neu, neutral; pai, pain; ple, pleasure; PPo, planum polare; R, right; pSTS, posterior superior temporal sulcus; sad, sadness; sem, standard error of the mean; STC, superior temporal cortex.

nature, we performed a last experiment (experiment 4) in which we recorded the brain activity of another sample of human listeners ($n$ = 30) using functional magnetic resonance imaging (fMRI) (Fig 3). We assumed that the human brain might process alarm screams with significantly greater efficiency compared with non-alarm screams as quantified by the level of neural signals in [1], and the connectivity between [26,31], brain areas that are central to affective sound processing. Voice signals and affect bursts are usually processed in a distributed brain network consisting of the auditory cortex, the amygdala, and the inferior frontal cortex (IFC) [1,26,32], which provide an acoustic and socio-affective analysis of these signals [33,34]. To

identify the neural dynamics of scream call processing in this network, we asked humans to listen to the same selected 84 screams as in experiments 2 and 3.

While listening to these scream calls, participants performed a largely orthogonal gender decision task on the screams, with orthogonal meaning that the task is largely independent from the emotional quality of the screams. The task was introduced to maintain the attention of the participants to the experiment, and is considered to yield implicit but still strong processing of the affective quality of the stimuli that leads to consistent brain activations [35,36]. RTs did not differ between the 7 scream types ($F_{6,174}$ = 2.371, $p$ = 0.094) nor between the 3 major categories of neutral, non-alarm, and alarm screams ($F_{2,58}$ = 0.769, $p$ = 0.414) (Fig 3A). The error rate was different between the 3 major categories of neutral, non-alarm, and alarm screams ($F_{2,58}$ = 3.747, $p$ = 0.041, but all post hoc comparisons with $p$ > 0.053), but showed a difference between all 7 scream types ($F_{6,174}$ = 7.280, $p$ < 0.001, $\eta^2$ = 0.10), with gender classification during pain screams showing higher accuracy compared to gender classification during anger, sadness, happy, and neutral screams (all post hoc $p$ < 0.030), and a lower gender classification accuracy during sadness compared to fear and pleasure screams (all post hoc $p$ < 0.018). Gender classification thus was equally performed in terms of speed and classification accuracy for the 3 major scream categories, but gender classification during the perception of certain scream types led to some specific better (pain) or worse performance (sadness) compared to other screams. This could be partly based on the acoustic profile of these screams (i.e., pain scream acoustics might obscure gender-related acoustic features less, while sadness screams might obscure this information more), and partly on the emotional relevance of pain and sadness screams (i.e., for physical pain, gender might be relevant to understanding the source of pain, while for "social pain" [sadness], gender becomes a minor relevant feature) [37,38].

While alarm screams (pain, anger, fear) mainly elicited lower neural activity in many inferior frontal and high-level auditory cortex regions, non-alarm screams (pleasure, sadness, joy) compared with neutral vocalizations showed higher and extended auditory cortical activations, especially in the right hemisphere in the low- and high-level auditory cortex (Fig 3D; S2 Fig; S1 Table; S2 Table). This pattern of neural activity is enhanced when the perceived valence of the screams is taken into account (S3 Fig). The activation in the auditory cortex largely extended across the voice-sensitive cortex, as determined by a separate voice localizer scan (Fig 3C). When we compared neural activity for alarming screams with that of non-alarming screams (Fig 3E), the non-alarming screams revealed extensive higher activity that was largely extended over the auditory and inferior frontal cortex. This indicates that a lower, not a higher, alarm level of screams is able to elicit more activity across many auditory and frontal brain regions; this was also shown in a separate analysis using the alarm rating across all screams as a parametric repressor, independent of the scream class (Fig 3D). Furthermore, neural activity for positive screams (pleasure, joy) as opposed to that for negative screams (sadness, pain, fear, anger) elicited similar largely extended activity in the auditory and inferior frontal cortex, as well as some activity in the bilateral amygdala (Fig 3F). Additional activity was found in the right posterior superior temporal sulcus (pSTS) for positive versus neutral vocalizations (Fig 3E).

To further examine the neural sensitivity for non-alarm screams, we performed an effective functional connectivity analysis on the observed patterns of brain activations using a dynamic causal modeling (DCM) approach. We sought to determine the neural information flow underlying the processing of alarm, non-alarm, and positive screams between the auditory cortex (sensory input and analysis), the IFC (cognitive evaluation), and the amygdala (affective analysis) in the left and right brain hemispheres (S4 Fig; S3 Table). Determining this neural information flow can provide a detailed picture about the integrated functioning and directed

interaction of diverse brain systems for decoding sociobiologically relevant information on sensory, cognitive, and affective levels. First, using DCM (Fig 3G and 3H), we found a bilateral basic (intrinsic) neural network with top-down and bottom-up connections from and to the frontal and auditory cortical subregions, respectively, as well as intrinsic connectivity between the posterior auditory cortex (pSTS) and the amygdala. This intrinsic network was independent of any experimental condition. Second, in terms of experimentally driven modulations in this network, we found significant driving input effects of positive screams in the right hemisphere, especially of non-alarm screams in both hemispheres, to the auditory cortical subregions. Third, non-alarm screams also modulated specific connections between auditory cortical subregions in both hemispheres, while positive screams modulated the connectivity from the posterior auditory cortex to the amygdala and the frontal cortex in the right hemisphere. No input, intrinsic, or modulatory connectivity effects were found for alarm screams in both hemispheres.

## Discussion

Our data provide evidence for a broad diversity of human scream calls with different affective meaning and communication profiles. The data from experiment 1 aimed at a broad psychoacoustic description of human scream calls, and these data go beyond the frequent assumption that scream calls are of a generic alarming nature [1,5,8,23]. Instead of scream calls being of a uniform acoustic and communication nature, related to threat and alarm signaling based on fear, we found several distinctive scream categories of alarming, non-alarming, and even positive nature in human primates. This distinctiveness was shown using a machine learning approach including an extensive set of relevant acoustic voice features [29]. Based on this approach, human scream calls appeared to acoustically differ significantly from other nonverbal affect vocalizations in a cross-vocalization comparison, but they were also distinctive across different types of screams when compared within the set of the 6 scream types. The acoustical patterns of screams thus seem to be distinct and largely specialized in the broad acoustic space of emotional vocalizations [6,25,26,30].

Besides this approach to compare scream calls based on a large acoustic feature set (i.e., 88 acoustic features), we also quantified another complex acoustic feature related to the spectral and temporal modulation rate of the acoustic signal, referred to as MPS [27,28]. We found that the spectral modulation rate showed high power across a broad frequency range, but the temporal modulation rates were not distributed across a broad frequency range (30–150 Hz) as previously reported [1]. Instead, we found that the temporal modulation rate was focused around 2 separate frequency bands centered on approximately 60 Hz and approximately 160 Hz, contributing to the harsh and rough acoustic quality of scream calls [39]. Furthermore, we also found that this roughness feature was not only characteristic of fearful scream calls as part of their alarming quality [1,23], but also of other alarm screams (pain, anger) and, most importantly, of some non-alarm screams (pleasure, sadness). Thus, while acoustic roughness seems to be an important feature of many scream calls, it is also critically related to the perceived alarm level of scream calls. Especially the low temporal frequency band of approximately 60 Hz of the acoustic roughness feature was related to the alarm level of scream calls, given the higher beta weight in the linear regression analysis for this frequency band to predict a scream's alarm level. The temporal modulation rate of approximately 60 Hz, and its range from 50 to 70 Hz, corresponds to the transition zone between acoustic flutter and a more continuous but rough pitch perception of sounds [40,41], with a potential sensitivity of the (primary) auditory cortex to such features [40,42,43].

In experiment 1 we demonstrated the psychoacoustic distinctiveness of various types of human scream calls of both alarming and non-alarming nature. In experiments 2 and 3 we

then turned to the question of the decisional processes that are involved while categorizing (i.e., into the 6 plus 1 scream categories) and discriminating different types of screams (i.e., discriminating 2 types of screams), especially alarm and non-alarm screams. Contrary to an assumed higher neurocognitive efficiency in processing alarm screams based on their proposed sociobiological relevance [1], alarm scream calls were overall responded to the slowest, were associated with a lower perceptual sensitivity, and were discriminated with a low accuracy in a 2-option decision task. There were some exceptions to this general observation that alarm screams were classified slower and with higher error rates. For example, sadness screams were classified slower when specifically compared to anger screams; furthermore, the performance for joy screams had large effects on the mean performance for non-alarm scream classifications. But when quantifying the $d'$ measure as an indicator of perceptual sensitivity as well as the false alarm rate, the general observation of a lower processing efficiency of alarm compared to non-alarm screams was largely confirmed. These findings are very surprising given that speed and accuracy in processing, as well as perceptual sensitivity, of the primate neurocognitive system is frequently assumed to be crucial for signals that indicate threat and danger [1,35,44–46]. Only in the speeded classification task—but not in the self-paced rating task that involved making ticks on a visual analog scale on the screen—we found indication for some lower processing efficiency of alarm compared non-alarm screams. This indicates that this lower processing efficiency for alarm screams only occurs when humans have to make fast judgments on the scream signals perceived, and these fast judgments seem critical in some contexts.

This lower efficiency in classifying alarm screams (with the exceptions mentioned above) was found not only in the more complex multi-option decision task of experiment 2, but also in the simpler 2-option decision task of experiment 3. We have to note that we used the exact same stimulus material in experiment 2 and 3 but, critically, changed the decisional task that participants had to perform. In experiment 3 we asked participants to decide between only 2 possible scream types, and found the highest RTs and error rates while discriminating between alarm screams. Furthermore, we found that discriminating alarm from non-alarm screams in some combinations also led to higher RT and errors for alarm screams. These results seem contrary to the usual assumption that discrimination of alarm screams, among themselves and from non-alarm screams, demands fast behavioral response and differential coping strategies, since we found the opposite pattern: that alarm screams took the longest to respond to and were discriminated with the lowest accuracy.

This additionally suggests that alarm screams, surprisingly, need more processing effort during simple scream discriminations with only 2 choice options in a speeded discrimination task, and that non-alarm and positive screams are more efficiently processed during this perceptual discrimination of 2 types of screams. Non-alarm screams and positive screams might be more relevant in social environments and interactions of human beings, and also might show a higher occurrence rate in daily lives [47]. The classification and discrimination of non-alarm and positive screams might thus have priority for humans and might explain their better processing efficiency. The only priority that alarm scream categories received was their frequent selection during misclassifications of other screams in the multi-option classification task in experiment 2. This might resemble a natural threat perception bias that seems to be a cost/benefit-efficient solution when balancing response options to potential sources of threat/nonthreat, especially under conditions of uncertainty [48]. Although perceptual misclassifications seem to be of general disadvantage to any receiving organism, these misclassifications as "alarm" screams might figure as a safer option in potential threat recognition. This bias might also be rooted in the evolutionary primacy of alarm screams to signal threat and elicit fear in receivers, signals that require immediate adaptive response in the organisms targeted by the

threat. Threat avoidance is thus the primary concern of an organism, and a "bias" towards incorrectly classifying non-alarm screams as threatening alarms is safer for an organism than performing the reverse error. One further specific note concerns the various combinations of 2 specific scream types in experiment 3. Some of these combinations might only rarely occur in daily life, but for completeness, we included all possible combinations here. For example, specific combinations of non-alarm and alarm screams might occur less frequently in daily life (e.g., joy and anger screams), but there are nonetheless certain contexts that include co-occurrences of non-alarm and alarm screams (e.g., a happy social gathering, where an interaction between 2 people suddenly turns violent because of an incident).

The behavioral data from the speeded classification tasks in experiments 2 and 3 thus seem to primarily point to a less efficient processing of alarm compared to non-alarm screams in these speeded classification contexts. This seems indicated by increased RTs when classifying screams (Fig 2A and 2E), a lower perceptual sensitivity when quantified with the $d'$ measure (Fig 2C and 2G), and higher RTs and lower discrimination accuracy when discriminating among alarm screams (Fig 2I and 2M). Also, alarm scream categories are more frequently chosen when humans misclassify screams (Fig 2D and 2H). We already discussed the latter observation above of being of some advantage for an organism, as this would be a safer option in case there is potential and suspected danger in the environment. Given the above description of the lower decisional efficiency in classifying alarm screams themselves, it might be that a critical function of alarm signals in general, and of alarm screams in specific, is to increase the alertness of the cognitive system to other ongoing events. Previous research has shown that alarm signals increase physiological responses in listeners [19,20], and that this facilitates detection of separate targets [21,22]. This focus on secondary targets might distract the attention away from alarm screams, and might lead to increased RTs for classifying the alarm scream itself and for discriminating it from other alarm screams.

In accordance with some decisional disadvantages when classifying and discriminating alarm compared to non-alarm screams, the neural patterns of functional activations in experiment 4 pointed to a sensitivity of the human neural system for non-alarm rather than for alarm screams. The neural processing of alarm screams largely resulted in decreased neural activations in comparison to other types of screams. Non-alarm and positive screams elicited increased neural activations in brain regions such as the IFC, the auditory cortex, and the amygdala. These regions are a core neural network that supports the social evaluation [49], acoustic analysis [31,33], and affective assessment of sounds [50,51]. Especially the auditory cortex and the amygdala were previously assumed to be reserved only for negative and alarming voice signals [1,31,52]. Using a parametric analysis approach, we also found that low, rather than high, levels of the alarm quality of screams were mainly driving the neural activity that we similarly found for comparing non-alarm and positive screams against other types of screams. Thus, the neural system seems not to be directly sensitive to alarm signals, at least when encoded in scream calls, but seems to be more sensitive to non-alarm and positive emotions. These neural effects resemble the behavioral decision-making profiles that we found in experiments 2 and 3.

Furthermore, modeling the directional neural connectivity between the frontal, auditory, and limbic brain regions, we also found that non-alarm screams, especially positive screams, show significantly greater effects in neural network dynamics over alarm screams, such that the human neural system is more prepared to decode non-alarming and positive information signaled in human scream calls in a mostly bottom-up manner. These findings in humans are surprising and largely diverge from studies in nonhuman primates. For the latter, the literature so far has only reported scream calls being expressed in negative contexts. While screams can also be expressed in negative contexts in humans, humans seem to be the only species to

express screams in non-alarm, and especially in positive, contexts. Concerning these non-alarm contexts and our neural connectivity analysis, we found that non-alarm screams provided significant input modulation to some auditory cortical subregions in both hemispheres and a modulation of a connection of auditory subregions in the right auditory cortex; positive screams showed significant input modulation in the right hemisphere as well as a modulation of the forward connection from the right posterior auditory cortex to both the right amygdala and the right IFC. Thus, there seem to be certain neural pathways connecting subregions in the right auditory cortex for non-alarm scream processing, as well as connecting the auditory cortex with the frontal cortex and the limbic system to decode positive meanings from screams. Previous work has shown connectivity from the auditory cortex to the amygdala [31] for acoustic information transfer to perform affective assessments, but only for unpleasant sounds. The auditory cortex also shows forward connection to the frontal cortex [53,54], but this forward connection was so far not shown to be modulated by the affective meaning of sounds.

Our data critically extend these previous findings, especially showing that these pathways in the right hemisphere are predominantly sensitive to positive voice calls. Most surprisingly again, alarm screams did not show any effects in this directional connectivity analysis, either as driving input to specific brain regions, as intrinsic connectivity, or as modulation of the connections between regions. We have to note that all neural activations, and specifically all connectivity results, were obtained while participants performed a non-emotional gender decision task, and thus participants were not explicitly focusing on the affective quality of the screams. This gender discrimination task might have also led to a relatively strong neural activity for neutral screams in comparison to some other types of screams, especially alarm screams. However, previous studies using a gender discrimination task during the processing of vocal emotions usually found higher activity for negative compared to neutral vocal emotions, and the implicit processing of the affective quality of voice signals has been demonstrated by many previous studies [35,36]. Given that the gender task in our study was largely unrelated to the affective and alarming quality of the scream types, we think that the neural activations and connected patterns largely reflect the neural dynamics for scream type processing and scream discrimination.

Before summarizing our main findings across the experiments reported here, we want to discuss a few potential limitations in our study. First, the scream vocalizations in our study were produced and acted by a sample of normal healthy humans who were, unlike professional actors or trained vocal speakers, relative naïve to the production of screams on command. It might be, therefore, that the acted screams in our study were not of full natural quality, like screams triggered by natural cues [2]. However, based on previous descriptions of what makes a scream, the screams produced by the naïve speakers in our study seemed to be of similar scream-like voice quality [2,25,55,56]. Furthermore, the perceptual assessment of the scream recordings by a sample of independent listeners confirmed that most of the screams could validly convey the intended emotional meaning and that they were of an alarming quality to a variable degree. Second, we found indications of some lower processing efficiency for alarm compared to non-alarm screams, especially in speeded classification tasks including complex 7AFC tasks as well as in a simpler 2-option discrimination task. No such differences were found in a self-paced and non-speeded rating task that asked humans to rate screams on their alarm level. The lower processing efficiency for alarm screams thus seems task-dependent and seems to be observed in contexts that require speeded decisions.

Taken together, this overall pattern of psychoacoustic, decision-making, and neural results for alarm screams in humans seems rather unusual for sociobiological voice calls when reviewed in the broader animal communication field. Scream calls were previously assumed to

be most crucial in signaling and accurate communication of alarm across a broad range of animal species [5,8,10], sometimes also including references to humans [1,23]. Scream calls in nonhuman primates and other animals have so far been reported to be expressed and perceived exclusively in negative contexts. A different picture of scream calls seems to emerge when investigated in humans, such that human listeners overall respond more quickly, more accurately, and with higher neural sensitivity to non-alarm and positive scream calls, which seem to have a higher relevance in human sociobiological interactions [57–59]. There seem to be some exceptions to this overall pattern of scream recognition in humans, but across the many psychoacoustic, behavioral, perceptual, and neural effects quantified here, alarm screams often show less neurocognitive processing efficiency than non-alarm screams. Alarm scream categories only have some primacy during misclassification of other scream types, which might be a safety choice under conditions of decisional uncertainty. And this safety choice might be shared with other nonhuman species that use screams in their vocal repertoire.

## Materials and methods

### Participants

All 4 experiments included an independent sample of human participants with normal hearing abilities and normal or corrected-to-normal vision. No participant presented a neurological or psychiatric history. All participants gave informed and written consent for their participation in accordance with the ethical and data security guidelines of the University of Zurich. The experiments were approved by the cantonal ethics committee of the Swiss canton Zurich (approval number KEK-ZH 2015–0329).

### Experiment 1: Acoustic dimension of scream calls

**Participants.**   In experiment 1, 12 healthy volunteers (6 males; mean age 29.08 years, SD = 5.66, age range 22–42) took part in the acoustic recording of 6 different types of screams, as well as a neutral vocalization (an intense utterance of the vowel /a/) as a seventh category. All participants were healthy humans with no acting experience, training in acting, or history of being professional actors.

**Experimental design.**   We invited participants to vocalize 6 different types of screams. These types of screams were chosen to cover a broad range of possible screams that humans vocalize in certain emotional states. A previous report on the acoustic and neural processing of screams identified screams as being largely only of a negative, fearful nature [1], to signal alarm to conspecifics [23], on the basis of a certain and unique acoustic feature of "roughness" (i.e., high-frequency spectro-temporal acoustic modulations). Although fearful screams are a prominent example of human screaming, humans produce vocal screams of a rough acoustic nature not only in the emotional state of fear, but also in a variety of emotional states referred to as "pleasure," "sadness," "joy," "pain," "fear," and "anger" states. These different types of screams can be classified as either positive (pleasure, joy) or negative (sadness, pain, fear, anger) screams, which we refer to as the factor "valence." Furthermore, screams can be classified as either alarm vocal signals (pain, fear, anger) or as non-alarm vocal signals (pleasure, sadness, joy), which we refer to as the factor "alarm quality." Thus, screams are not only limited to negative and alarm screams of fear, but they can also be positive and of a non-alarm nature.

We thus instructed the 12 participants to produce vocalizations of screams for each of the 6 types of screams, as well as to produce neutral screams on the basis of an intense vocalization of the vowel /a/. For each type of scream, we provided 2 short instructions to each participant to imagine exemplary contexts in which these screams are commonly produced (e.g., fearful

scream: "You are being attacked by an armed stranger in a dark alley"; anger scream: "You try to intimidate an opponent"; joyful scream: "Your favorite team wins the World Cup"; pleasure scream: "You are screaming from sexual delight"). These short instructions were provided such that participants would have 2 example situations in which each scream type is typical (but not exclusively) expressed. This type of procedure with short instructions and examples is similar to previous research in the field [60]. Participants were instructed to imagine the emotional quality of some of the provided contexts, or similar contexts of the same emotional quality, and produce screams that would reflect their own vocal expression according to the emotional quality of this imagined context, but not necessarily of the specific person and/or event described in the short instruction. Before the actual recording, participants did a training session of approximately 15 min with the experimenter to ensure that the emotional context for each scream was understood correctly and that appropriate intensive vocalizations of a scream-like character (e.g., a harsh, loud, rough, and thrilling sound quality of the voice, as described in the literature [1,55,56,61]) could be produced for each emotional context. Participants were instructed to deeply inhale before each vocalization to enable maximum glottal pressure for each single vocalization. Using these instructions and this training, we recorded 8 instances of each type of scream for each speaker in an anechoic chamber with a Neumann TLM-102 microphone at a distance of approximately 1 m from the speaker. From these 8 instances, we chose the 5 best recordings based on a perceptual judgment of recording quality, vocalization length (800–900 ms), continuous vocalization for the duration of the scream, and the perceptual impression of having a scream-like vocalization quality.

The final selected screams were cropped to a fixed duration of 800 ms, standardized to an identical root mean squared (RMS) signal across screams corresponding to 70 dB SPL, and faded in and faded out by a 15-ms intensity ramp at the beginning and end of each scream. The final sample included 420 screams. Although this standardization of the screams to 800 ms and 70 dB SPL might render some of the screams partly unnatural (e.g., alarm screams might have a higher natural loudness/intensity than non-alarm screams), this procedure ensured that all following analyses and perceptual experiments are not confounded by basic acoustic features of these screams due to vocalization characteristics and recoding conditions for single speakers. To avoid experimental confounds, we considered the exact standardization of the stimuli to be of higher importance for a straightforward interpretation of the stimuli compared to introducing a little bit less natural sounding screams. Furthermore, although these screams were largely acted rather than spontaneously expressed, acted screams seem to be perceptually similar to natural screams [1,2], and thus provide a valid basis for affective communication research in human and nonhuman primates.

**Perceptual assessment of screams.** After the acoustic recording of the screams, we asked 23 participants (12 males; mean age 24.04 years, SD = 5.35, age range 20–46) to perceptually assess the alarm quality of each of the 420 screams on a continuous scale ranging from 0 (not alarming at all) to 1 (highly alarming). This task was performed by positioning a slider on a visual analog scale presented on the screen, and moving the slider to the appropriate rating point. Next, we asked another sample of 26 participants (11 males; mean age 24.40 years, SD = 3.56, age range 19–32) to perceptually assess these screams according to the central emotional dimensions of arousal and emotional category. Arousal was quantified on a 7-point Likert scale ranging from 1 ("not arousing at all") to 7 ("highly arousing"); emotional category was quantified by a 7AFC classification task (neutral, pleasure, sadness, joy, pain, fear, anger). Based on this perceptual assessment and the recognition accuracy for the screams, we selected 84 screams from 3 male and 3 female speakers (2 instances per scream type for each speaker) for the subsequent experiments (see below). Both the alarm rating and the perceptual assessment acquisition were performed in a self-paced manner, such that participants could take

their own time to perform the ratings; the next trial only started after the participants finished the ratings on the previous trial.

**Data analysis.** The data from the perceptual assessment were separately analyzed for the arousal ratings and the classification ratings. Arousal ratings were subjected to a repeated-measures 1w-ANOVA that included the within-participant factor scream type (levels: neutral, pleasure, sadness, joy, pain, fear, anger). Using a similar ANOVA, we also analyzed the classification accuracy of the screams. In addition, we quantified the confusion matrix (i.e., individual percentage of categories chosen during misclassification of each type of scream) and the false alarm rate (i.e., the probability of a category being chosen during misclassifications). The significance threshold was set to $p = 0.05$ (here and for all analyses that follow).

The alarm ratings were analyzed in a similar way as for arousal and valence ratings. For an additional analysis of the alarm ratings, we used full permutation and statistical testing through all possible categorical combinations of the 6 scream types, with the described categorization (pain, fear, and anger as alarm screams and pleasure, sadness, and joy as non-alarm screams) leading to the most significant distinction in alarm level between screams. During this permutation approach, 1 category was always represented by neutral screams, while the 6 scream types were randomly split into 2 separate categories of screams with 2–4 scream types per category. This led to 25 possible combinations of screams. For each combination, we calculated the mean alarm rating for each participant and subjected these data to a repeated-measures 1w-ANOVA with 3 levels.

**Acoustic analysis.** The acoustic analysis of screams and other vocalizations was based on 88 previously described acoustic features [29] extracted with the toolbox openSMILE [62]. The acoustic feature set comprised frequency (pitch, jitter, etc.), energy/amplitude (intensity, shimmer, etc.), and spectrum-related features (alpha ratio, Hammarberg index, etc.), as well as 6 features related to the temporal domain (rate of loudness peaks, voiced segments, etc.). Arithmetic mean and coefficient of variation (standard deviation normalized by the arithmetic mean) were returned for all parameters. For the loudness parameter, the 20th, 50th, and 80th percentiles; the range of 20th to 80th percentiles; and the mean and standard deviation of the slope of rising/falling signal parts were also computed. Each acoustic parameter was normalized (i.e., centered on the mean and divided by its standard deviation) across all 420 sounds.

Besides these basic acoustic features, we also analyzed the MPS [28] using the MPS toolbox for MATLAB (https://github.com/theunissenlab/soundsig). To obtain an MPS for each sound, we first converted the amplitude waveform to a log amplitude of its spectrogram obtained using Gaussian windows and a log-frequency axis. The MPS results from the amplitude squared as a function of the Fourier pairs of the time (i.e., temporal modulation in hertz) and frequency axis (i.e., spectral modulation in cycles/octave) of the spectrogram. The low-pass filter boundaries of the modulation spectrum were set to 200 Hz for the temporal modulation rate and to 12 cycles/octave for the spectral modulation rate. A statistical difference of the MPS of scream types compared to neutral screams was tested using a permutation approach ($n$ = 2,000) by shuffling scream type labels, resulting in a $p$-value map for the entire spectral and temporal range of the MPS (Fig 1B).

**Machine learning approach.** Using the 88 acoustic features that were extracted from each scream call using the openSMILE toolbox as described above, we used a machine learning approach to assess the acoustic distinctiveness of the various types of screams. For each scream category and the neutral vocalizations, we trained an SVM [63] to discriminate these 7 emotion categories of screams on the basis of all acoustic parameters as features (see [64] for a discussion of classification methods). This approach is called a "cross-validation" approach. The SVM is a binary classifier, and so multiclass classification can be set up in either a 1-versus-1 or a 1-versus-all scheme [65]. We used a 1-versus-1 scheme as implemented by MATLAB's

fitecoc function (MATLAB, version 2018a), which estimates a classification model for each available pair of vocalization categories separately. The SVM parameters were set to the MATLAB default values, and the kernel function was a third-order polynomial.

To evaluate the classification accuracy for the scream discrimination in the cross-validation approach, we tested the SVM on unseen testing data that were not used to estimate the model parameters. Within each scream category, a 5-fold cross-validation scheme split the sound set (420 sounds, with 60 sounds per emotion) repeatedly into a training and a test dataset, such that each sound served once as a test data point. This approach estimates how well the SVM is capable of separating the 7 scream categories (6 scream calls, 1 neutral vocalization) based on their acoustic features.

Next, a new SVM model learned to separate all scream types and predicted the emotion label on another set of more standard nonverbal expressions of emotion (e.g., joyful laughter and anger growls) of the same valence as the scream types and vice versa. The nonverbal vocal expressions consisted of 448 emotional voices from 8 male and 8 female speakers (mean age 30.38 years, SD = 7.07, age range 21–47; 10 of these 16 speakers also vocalized the scream calls used here) that included the vocal expressions neutral (vowel /a/ in a normal voice), pleasure, sadness, joy, pain, fear, and anger. These nonverbal vocal expressions were recorded with the same kind of settings that we used to record the scream calls. From every speaker, we had 4 instances of a vocalization for each emotion. This approach is referred to as a "cross-classification" approach, and tests whether a model learned a representation of the emotion categories that generalizes across vocalization categories (screams, nonverbal expressions).

## Experiment 2: Perceptual categorization of scream calls

**Participants.** In experiment 2, 33 healthy volunteers (12 males; mean age 23.91 years, SD = 3.28, age range 19–33) took part in the experiment on the perceptual categorization of 7 different types of screams (including the neutral vocalization). We replicated experiment 2 with another independent sample of 29 healthy volunteers (18 males; mean age 25.96 years, SD = 4.30, age range 18–32), but using a different selection of scream stimuli.

**Experimental design.** From the acoustic scream recordings of experiment 1, we selected 84 screams (3 male and 3 female speakers) with 2 instances of screams per speaker per category. Stimuli were selected from the results of the perceptual assessment of screams in experiment 1, such that no significant differences in the recognition rate across scream types ($F_{6,150}$ = 1.895, $p$ = 0.117) were found for this selection. Mean arousal level differed across all 7 scream types ($F_{6,150}$ = 51.065, $p < 0.001$), across the 6 scream types ($F_{5,125}$ = 11.647, $p < 0.001$), and between neutral, alarm, and non-alarm screams ($F_{2,50}$ = 84.558, $p < 0.001$). The experiment consisted of 3 sessions. One session included 168 trials, and during each session the 84 screams were presented twice. Screams were presented at 70 dB SPL using high-quality headphones with a flat frequency profile (Sennheiser HAD 280), and participants were asked to perform a 7AFC task according to the emotional valence of the screams (neutral, pleasure, sadness, joy, pain, fear, anger). Decision options were presented on the screen in a circular arrangement, each at approximately 5˚ visual angle from the center of the screen. Participants had to respond by moving the mouse cursor from the middle of the circle (i.e., mouse cursor position was reset to the middle of the circle after each trial) to 1 of the response options and pressing the left mouse button with the right index finger. Participants were asked to respond as quickly as possible, while still trying to make an accurate decision. The arrangement of response options was randomly chosen for each participant but was kept constant across blocks for each participant. Screams were presented in random order at 70 dB SPL, with a maximum RT window of 3,000 ms; trials with no response in this time window were classified as missed trials (6.53% of

all trials, range 4.41%–7.74% across scream types). After a response option was chosen, the response screen disappeared and the next scream was presented after an inter-trial interval (ITI) of 1,750 ± 250 ms. Participants could give their response in a 3-s time window; if no response was given within these 3 s, the experiment continued with the next trial.

As mentioned above, we repeated the same experiment with another sample of participants, but using a different selection of 84 stimuli out of the total 420 stimuli. This replication was done to show that the selection of a certain subsample of screams did not affect the general pattern of results obtained in experiment 2. Out of the 420 screams we thus selected another sample of 84 screams that overlapped with the original selection by 25%. This overlap was due to the fact that the selection procedure was guided by the constraint that the selected screams should not have a difference in their base recognition rate. Out of the total number of $n = 9.1626 \times 10^{89}$ possible solutions for selecting 84 stimuli out of 420 screams, we ran $n = 10,000,000$ simulations, and it was not possible to obtain a new selection of $n = 84$ stimuli that did not include some of the previously selected stimuli while maintaining the restriction that the base recognition rate not be significantly different between the 7 stimulus categories. The minimal solution was to allow an overlap of selected screams of approximately 25% between the first and the second selection. The second selection of 84 screams again did not show significant differences in the recognition rate across scream types ($F_{6,150} = 1.596$, $p = 0.182$). Mean arousal level differed across all 7 scream types ($F_{6,150} = 45.137$, $p < 0.001$), across the 6 scream types ($F_{5,125} = 10.551$, $p < 0.001$), and between neutral, alarm, and non-alarm screams ($F_{2,50} = 72.327$, $p < 0.001$).

**Data analysis.**   RT and classification accuracy data were separately subjected to a repeated-measures 1w-ANOVA that included the within-participant factor scream type (levels: neutral, pleasure, sadness, joy, pain, fear, anger). RTs were defined as the response latency from stimulus onset to the button press, and RTs were only calculated for correct trials. Accuracy was defined as pressing the 1 button that corresponded to the preassigned category of a scream (e.g., pressing the button for "fear" when a fear scream was presented). We performed an additional 1w-ANOVA that included 3 categories: neutral, alarm, and non-alarm screams. We again quantified the confusion matrix and the false alarm rate. From the correct classification for a scream category and the false alarm rate that involved a certain scream category, we calculated the $d'$ measure as an indicator of the sensitivity and detectability of a certain scream type. The $d'$ measure was calculated according to the formula $d' = Z$(hit rate) $- Z$(false alarm rate). The $d'$ measures were subjected to the same ANOVA analyses as described above.

### Experiment 3: Perceptual discrimination of scream calls

**Participants.**   Thirty-five healthy volunteers (11 males; mean age 24.34 years, SD = 3.25, age range 20–30) took part in the experiment on the perceptual discrimination of screams.

**Experimental design.**   The same 84 screams as in experiment 2 were used. Screams were presented in 21 blocks, each block containing 2 different scream types. Over the 21 blocks, we presented all possible combinations of the 7 scream types. One block consisted of 72 trials. For example, in 1 block, we presented only angry (36 trials) and fearful (36 trials) screams. Screams were presented as single trials in a random order, and after each scream participants performed a 2-alternative forced-choice (2AFC) task by pressing 1 button (left index finger) for 1 scream category (e.g., "anger") and another button (right index finger) for another scream category (e.g., "fear"). Participants were again asked to respond as quickly as possible, while still trying to make an accurate decision. Trials with no response in this time window were classified as missed trials (1.55% of all trials, range 0.39%–3.32%). In each block, screams were presented in random order at 70 dB SPL, with an ITI of 2,250 ± 250 ms. Button and response option assignments were counterbalanced across blocks and participants.

**Data analysis.** For each of the 21 combinations of screams, we scored the mean RTs (i.e., response latency from stimulus onset to button press, only for correct trials) and accuracy (i.e., correct trials were defined as a button press that corresponded to the preassigned scream type) for each of the 2 scream categories, as well as the mean difference in RTs and accuracy between the 2 categories for each participant. The mean difference values for each scream combination were normalized by using the mean RT (1/mean RT) or the mean accuracy rate (mean percent) for the 2 scream types for each participant. For each of the 21 scream combinations, we tested for significant differences in RTs and accuracy using a $t$ test; the significance threshold was adjusted for multiple testing by using the FDR correction. We performed an additional 1w-ANOVA comparing mean RTs and accuracy rates for 4 categories of potential scream combinations: neutral screams combined with category screams (e.g., neutral screams combined with fear screams), non-alarm screams combined with non-alarm screams (e.g., pleasure screams combined with sadness screams), non-alarm screams combined with alarm screams (e.g., pleasure screams combined with anger screams), and alarm screams combined with alarm screams (e.g., fear screams combined with anger screams).

## Experiment 4: Neural processing of scream calls

**Participants.** We invited 30 healthy volunteers to take part in experiment 4 (15 males; mean age 26.10 years, SD = 4.95, age range 18–43). Participants were informed about the aim to investigate the neural dynamics of processing different types of screams, and they were informed to perform a task that is orthogonal to the emotional dimension of screams.

**Experimental design.** During the fMRI experiment, we presented the same 84 screams as in experiment 2. The experiment included 2 blocks, and each block included 168 trials. For each block, we acquired 565 functional images for each participant. Screams were presented in random order at 70 dB SPL, with an ITI of 3,875 ± 875 ms. For the presentation of the stimuli, we used active noise-cancellation MR-compatible headphones (OptoActive II; https://www.optoacoustics.com) that can reduce MRI scanner noise by up to 20 dB. Participants performed a gender decision task (Fig 3A), which was a task that was orthogonal to the relevant dimension of processing the different types of screams. This gender task has been used in previous studies and was established to be validly orthogonal to the affective dimension of stimuli [32,35]. After each scream presentation, participants indicated the corresponding gender of the speaker by pressing 1 button (right index finger) when the speaker was "male" or the other button (right middle finger) when the speaker was "female." Trials with no response in this time window were classified as missed trials (1.87% of all trials, range 0.69%–3.04%). This basic task was implemented to keep the participants attentive throughout the experiment. Participants were again asked to respond as quickly as possible, while still trying to make an accurate decision. The task was orthogonal to the affective valence dimension of screams, which was the target dimension for the main analysis. Response button assignment was counterbalanced across participants but was constant across blocks for a single participant. Functional brain data were recorded on a Philips Ingenia 3T using a standard 32-channel head coil. Data preprocessing was done according to standard principles including normalization to the Montreal Neurological Institute (MNI) stereotactic space.

For the statistical analysis of the functional brain data, each experimental condition was modeled as a separate regressor in a participant-wise GLM analysis, and the resulting images were taken to a second-level group analysis. All functional group results were thresholded at a combined voxel threshold of $p < 0.005$ corrected for multiple comparisons at a cluster level of $k = 42$. Modeling the effective connectivity between activated brain areas based on the various contrasts performed was done using DCM as implemented in SPM12 [66], including a

stepwise procedure. We first created 3 different families of models (forward, backward, and bidirectional models) for each hemisphere separately on the basis of the definition of the input to regions (C matrix), the intrinsic connection between regions (A matrix), and the modulation of connection by experimental conditions (B matrix) (S4 Fig). After estimating the winning model family in each hemisphere, we used a Bayesian model averaging (BMA) approach, which creates a weighted average of all models within the winning family by taking the model evidence of each model into account. The resulting posterior estimates for each parameter in the A, B, and C matrices and the weighted average model were tested for significance using a *t* test against "0," with resulting *p*-values adjusted for multiple comparisons using the FDR correction.

**Voice localizer scan.**   The stimulus material consisted of 500-ms sound clips [67] consisting of 70 human speech and non-speech vocalizations and of 70 nonhuman vocalizations and sounds (animal vocalizations, artificial sounds, natural sounds) presented at 70 dB SPL. Each sound was preceded by a 500-ms fixation cross and followed by a jittered blank 3,550- to 5,000-ms gap before the onset of the next stimulus. Each sound clip was presented once during the experiment. Participants were asked to perform a 2AFC task (right hand: index and middle finger; button assignment counterbalanced across participants) on the stimuli to decide whether they heard a vocal or a nonvocal sound. The experiment was preceded by 10 training trials (random selection of 5 vocal and 5 nonvocal trials) to familiarize the participants with the task. The training trials were not included in analyses. For the voice localizer scan, we acquired a total of 470 functional images for each participant (Fig 3C).

**Image acquisition.**   The recording of structural and functional brain data was done on a Philips Ingenia 3T using a standard 32-channel head coil. High-resolution structural MRI was acquired using T1-weighted scans (TR 7.91 ms, TE 3.71 ms; voxel size 0.57 mm$^3$, in-plane resolution 256 × 251 voxels). Functional whole-brain images were recorded with a T2$^*$-weighted echo-planar pulse sequence (TR 1.65 s, TE 30 ms, FA 88˚; in-plane resolution 128 × 128 voxels, voxel size 1.71 × 1.71 × 3.5 mm; gap 0.4 mm; 19 slices) with particle volume acquisition of slices rotated approximately 30˚ nose-up to the anterior commissure–posterior commissure (AC–PC) plane. The partial volume covered the superior temporal cortex (STC), IFC, and medial limbic system (i.e., the amygdala). Data preprocessing was done according to standard pipeline including normalization to the MNI stereotactic space and isotropic smoothing (6 mm) of the functional data.

**Data preprocessing.**   Preprocessing and statistical analyses of functional images were performed with Statistical Parametric Mapping software (SPM12, Welcome Department of Cognitive Neurology, London; https://www.fil.ion.ucl.ac.uk/spm). Functional data were first manually realigned to the AC–PC axis, followed by motion correction of the functional images. Each participant's structural image was co-registered to the mean functional image and then segmented to allow estimation of normalization parameters. Using the resulting parameters, we spatially normalized the anatomical and functional images to the MNI stereotactic space. The functional images for the main experiment were resampled into 1.7-mm$^3$ voxels. All functional images were spatially smoothed with a 6-mm full-width half-maximum isotropic Gaussian kernel.

**Single-participant and group analysis.**   For the first-level analysis of the main experiment and the voice localizer scan, we used a general linear model (GLM), and all trials were modeled with a stick function aligned to the onset of each stimulus, which was then convolved with a standard hemodynamic response function. For the main experiment, trials were modeled for each of the 7 scream types by separate regressors in 1 GLM, and for each trial, we entered the arousal level of each scream from the pre-evaluation results into the GLM as a covariate of no interest. The scream types differ in their arousal level, which might confound the neural brain

data. For the voice localizer scan, we modeled trials with vocal and nonvocal sounds separately in another GLM. For each GLM of the main experiment and the voice localizer scan, we also included 6 motion correction parameters as regressors of no interest to account for signal changes not related to the conditions of interest.

Contrast images for the main experiment were then taken to a random-effects group-level analysis to investigate the neural processing of the different types of alarm and non-alarm screams by performing several directed contrasts between conditions (S2 Fig). Contrast images for the voice localizer scan were then taken to separate random-effects group-level analyses to determine voice-sensitive regions in both hemispheres of the auditory cortex. All group results were thresholded at a combined voxel threshold of $p < 0.005$ corrected for multiple comparisons at a cluster level of $k = 42$. This combined voxel and cluster threshold corresponds to $p = 0.05$ corrected at the cluster level and was determined by the 3dClustSim algorithm implemented in the AFNI software (https://afni.nimh.nih.gov/afni; version AFNI_18.3.01; including the new [spatial] autocorrelation function [ACF] extension) according to the estimated smoothness of the data across all contrasts. The cluster extent threshold of $k = 42$ was the maximum value for the minimum cluster size across contrasts of the main experiment and the functional voice localizer scans.

For the main experiment, after the GLM analysis with predefined scream categories based on the original scream recordings and scream selection (Fig 3D–3F), we performed the same analysis using the subjective post-experimental scream classification of each participant (S3 Fig). In this analysis, trials for the 7 scream categories were defined from the classification of each participant, e.g., an original anger scream that was classified as a "sadness" scream by a participant was used as a sad scream trial in the analysis. Thus, this analysis was based on the perceptual impression of each participant rather than on the preassigned scream classification. This analysis must, however, be interpreted with caution. First, it resulted in an unequal number of trials for each category and for each participant, since this depended on the classification performance of each participant. Second, the analysis is not sensitive to the classification errors of participants. Some of the trials might thus include error trials, where participants subjectively pressed the wrong button but recognized their error. Third, these scream classifications were performed after the experiment and not during the fMRI experiment. However, overall, this analysis might give a good estimate of the neural activity that underlies the perceptual impression of the participants.

We performed an additional parametric modulation analysis by modeling each trial on a single-participant level using the alarm ratings for each scream as a covariate, including 1 regressor for all trials and an additional regressor for the covariate in a GLM analysis. The arousal level was again included as a covariate of no interest. Single-participant beta images were taken to a second-level group analysis as described earlier. Contrasts were computed for positive and negative associations with the alarm ratings as covariates of no interest (Fig 3B).

**DCM of effective brain connectivity.** We used the DCM12 toolbox implemented in SPM12 to model the information flow in the left and right brain hemispheres during the processing of different types of screams. We defined a set of regions of interest (ROIs) in the auditory cortex, amygdala, and IFC on the basis of group peak activations for the different contrasts performed. For each definition of a peak, the contrast was chosen that was the most informative about the potential sensitivity of the region for certain experimental conditions and scream types. The amygdala peaks (left [−29 2 −17], right [30 −3 −17]) were based on the contrast [positive > negative], and the IFC peaks (left [−44 10 29], right [42 9 27]) were based on the contrast [non-alarm > alarming]. In the left hemisphere, we furthermore defined peaks in the planum polare (PPo; [−42 −7 −21], contrast [non-alarm > neutral]), middle superior temporal gyrus (mSTG; [−61 −25 5], contrast [neutral > alarm]), and pSTS ([−63 −−35 8],

contrast [non-alarm > alarm]). In the right hemisphere, we defined the pSTS ([47 −37 5], contrast [positive > neutral]), mSTG ([63 −13 0], contrast [non-alarm > neutral]), and middle superior temporal sulcus (mSTS, [54 −24 −2], contrast [neutral > alarm]). Signal time courses from these ROIs were extracted for each participant in a sphere with a radius of 3.4 mm, including only supra-threshold voxels ($p < 0.05$). The signal time course was quantified as the first eigenvariate representing the most typical time course across voxels included in each ROI. Acquisition delay for each ROI was accounted for by estimating the time of signal acquisition in each ROI across the time interval of the repetition time (TR).

We created 3 different families of models for each hemisphere separately on the basis of the definition of the input to regions (C matrix), the intrinsic connection between regions (A matrix), and the modulation of connection by experimental conditions (B matrix). The first model family is referred to as "forward models" ($n = 128$ models for the left hemisphere and $n = 64$ models for the right hemisphere) from the definition of the A matrix, including the bidirectional connections between regions in the auditory cortex, but only forward connections to the amygdala and the IFC. The family of "backward models" ($n = 128$ for left and $n = 64$ for right) is identical to the family of forward models, but only includes a backward connection from the amygdala and the IFC to the STC regions. The "bidirectional models" ($n = 4,096$ for left and $n = 1,028$ for right) include bidirectional connections between the amygdala and the IFC on the one side, and between the amygdala and the STC regions on the other side. For the left hemisphere, the input C matrix included non-alarm trials to the pSTS and the PPo, as well as alarm trials to the mSTG, while for the right hemisphere, the C matrix included positive trials to the pSTS, non-alarm trials to the mSTG, and alarm trials to the mSTS. This input C matrix was fixed for all models and only included subregions of the auditory cortex as potential regions for experimental input, but we estimated the strength and significance of these driving inputs for each model. Concerning the B matrix, we defined connectivity modulations from the predominant response of the seed and target region; for example, the connectivity to and from the amygdala was supposed to be only modulated for positive screams, since activation in the amygdala was mainly driven by those screams. For details about the B matrix specification, as well as the specification of the A and C matrices, see S3 Fig.

For each model family and each hemisphere, we estimated DCM for any possible combination of parameters in the B matrix, ranging from no modulation of connectivity to full modulation of any connection as defined in the A matrix. For the bidirectional model family in the left hemisphere, this, for example, resulted in a total of $n = 4,096$ models. To define the winning model in each hemisphere, we used a 2-stage approach. First, we compared model families within each hemisphere and defined the winning family on the basis of a Bayesian model selection using a fixed-effects approach, assuming the same model architecture across the participants [68], which is a similar approach to that in a previous study on auditory processing [69]. Second, within the winning family, we used a BMA approach, which creates a weighted average of all models within the winning family by taking the model evidence of each model into account. The resulting posterior estimates for each parameter in the A, B, and C matrices and the weighted average model were tested for significance using a $t$ test against "0," with resulting $p$-values adjusted for multiple comparisons using the FDR correction. For DCM in the left hemisphere, we had to exclude $n = 2$ participants, and for right hemispheric DCM, we had to exclude $n = 2$ participants, given extreme values for several of the estimated parameters [69].

**Post-experimental perceptual assessment of screams.** After the fMRI experiment, we asked every participant to perform a 7AFC task according to the emotional category of the screams (neutral, pleasure, sadness, joy, pain, fear, anger), similar to that for experiment 2. This was done outside the scanner because this rather complex categorization task with 7

response options was difficult to implement inside the fMRI environment. Before each emotional classification, we asked participants to rate the arousal level of each scream on a 7-point Likert scale ranging from 1 ("not arousing at all") to 7 ("highly arousing").

Classification accuracy data were separately subjected to a repeated-measures 1w-ANOVA that included the within-participant factor scream type (levels: neutral, pleasure, sadness, joy, pain, fear, anger). We again quantified the confusion matrix and the false alarm rate.

## Supporting information

**S1 Data. Excel spreadsheet containing, in separate sheets, the underlying numerical data for Fig 1C–1F.**
(XLSX)

**S2 Data. Excel spreadsheet containing, in separate sheets, the underlying numerical data for Fig 2A–2M.**
(XLSX)

**S3 Data. Excel spreadsheet containing, in separate sheets, the underlying numerical data for Fig 3A, 3G and 3H.**
(XLSX)

**S4 Data. Excel spreadsheet containing, in separate sheets, the underlying numerical data for S1A–S1C Fig.**
(XLSX)

**S1 Fig. Affective evaluation of the 6 scream calls and the "neutral" screams.** (a) Overall rating of the total $n = 420$ screams by $n = 26$ participants with respect to the recognition rate (top), their confusion matrix (middle top), the false alarm rate quantifying which categories were used during misclassifications (middle bottom), and the arousal ratings of each scream type (bottom). (b) The same measure for the selected $n = 84$ screams that were used in the behavioral and the fMRI experiments. (c) Participants in the fMRI experiment also performed a post-experimental rating of the same selected screams as in (b). Numerical data underlying the plots (a–c) can be found in S4 Data; see S1 Text for statistical analyses.
(TIF)

**S2 Fig. Brain activations for comparing single scream types with neutral screams.** Functional activation for contrasting each of the 6 scream types against neutral screams ($n = 30$). This resulted in either "positive" activations (left 2 columns: higher activity compared with neutral screams) or "negative" activations (deactivations) (right 2 columns: lower activity compared with neutral screams). Threshold $p = 0.005$ voxel level, cluster size of $k = 42$ (corrected $p = 0.05$ at cluster level).
(TIF)

**S3 Fig. Functional activations based on classification results of the post-experimental rating.** (a) Functional activations for alarm and non-alarm screams compared with neutral screams, with trials sorted on the basis of how participants ($n = 30$) classified each scream in the post-experimental rating. The analysis included 7 types of screams similar to those in the original analysis in Fig 3. (b) Functional activations based on comparisons within the scream types from post-experimental classifications. (c) Functional activations for positive screams compared with neutral and negative screams. Threshold $p = 0.005$ voxel level, cluster size of $k = 42$ (corrected $p = 0.05$ at cluster level).
(TIF)

**S4 Fig. Specification of the model space for DCM.** Three model families were created for both the left and the right hemisphere, including a family of forward models (left), a family of backward models (middle), and a family of bidirectional models (right). While all families were identical on the local connection of subregions in the auditory cortex, they differed in terms of connections from auditory subregions to the amygdala and the IFC. Forward models included connections from the auditory cortex to the IFC and the amygdala, backward models included connections from the IFC and the amygdala to the auditory cortex, and bidirectional models included bidirectional connections between the auditory regions and the amygdala and IFC. These connections defined the A matrix for DCM and were constant across the permutation across the model space derived from the B matrix. The B matrix defined the modulation connection by experimental conditions, which are color coded in the figure. The alarm condition (red) included only trials with alarm screams (pain, fear, anger), the non-alarm condition (orange) included only trials with non-alarm screams (pleasure, sadness, joy), and the positive condition (green) included only positive screams (pleasure, joy). The input C matrix (small arrows) defined the condition that provides input into the connectivity matrix, and we defined the input to each of the auditory subregions on the basis of predominant activity (or deactivations) resulting from the group-level contrasts.
(TIF)

**S1 Table. Peak activations for group-level contrasts between scream categories.** The table lists all significant peak activations for contrasting scream categories against each other (a–n). All activations are threshold at $p < 0.05$ corrected at the cluster level.
(PDF)

**S2 Table. Group-level peak activations for the parametric analysis including alarm ratings.** (a) No functional activations were found for a positive parametric analysis, taking the rated alarm level of each scream into account. (b) Functional activations for a negative parametric analysis with the alarm level of each scream. All activations are threshold at $p < 0.05$ corrected at the cluster level.
(PDF)

**S3 Table. Estimated connectivity parameters resulting from the DCM analysis.** (a) Group parameter estimates (FDR corrected $p$-values in parentheses) resulting from Bayesian model averaging (BMA) of the estimated parameters for the C matrix (input matrix), the A matrix (intrinsic connectivity matrix), and the B matrix (connection modulation matrix) for the left hemispheric models of the bidirectional family (winning family). The left hemisphere dynamic causal models included 5 regions; for the A and B matrix, the table is organized such that columns are the origin of the connection and rows are the target of the connection. Cell entries denoted as "na" were connections in the A matrix, but since for the left hemisphere, a single model was the overall winner, BMA was applied only to this model that had 2 B matrix entries that were not part of the estimation process. (b) Group parameters for the right hemisphere.
(PDF)

**S1 Text. Statistical analysis for data presented in S1 Fig.**
(PDF)

## Acknowledgments

We thank Plamina Dimanova for help during data acquisition, and we thank Claudia Roswandowitz, Simon Townsend, and Dominik Bach for helpful comments on the manuscript.

## Author Contributions

**Conceptualization:** Sascha Frühholz, Wiebke Trost.

**Data curation:** Sascha Frühholz, Joris Dietziker, Matthias Staib.

**Formal analysis:** Sascha Frühholz.

**Funding acquisition:** Sascha Frühholz.

**Investigation:** Sascha Frühholz.

**Methodology:** Sascha Frühholz, Wiebke Trost.

**Project administration:** Sascha Frühholz.

**Visualization:** Sascha Frühholz.

**Writing – original draft:** Sascha Frühholz, Wiebke Trost.

**Writing – review & editing:** Sascha Frühholz, Wiebke Trost.

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
