## [Editor Report · Decision Letter 0]

28 Apr 2020

Dear Dr Frühholz, 

Thank you for submitting your revised manuscript entitled "Neurocognitive processing efficiency for non-alarm rather than alarm signaling in human scream calls" for consideration as a Short Reports by PLOS Biology.

Your revisions have now been evaluated by the PLOS Biology editorial staff, and I'm writing to let you know that we would like to send your submission out for re-review.

Please re-submit your manuscript within two working days, i.e. by Apr 30 2020 11:59PM.

Kind regards,

Roli Roberts

Senior Editor

PLOS Biology

---

## [Decision Letter · Decision Letter 1]

13 Aug 2020

Dear Dr Frühholz,

Thank you very much for submitting a revised version of your manuscript "Neurocognitive processing efficiency for non-alarm rather than alarm signaling in human scream calls" for consideration as a Short Reports at PLOS Biology. This revised version of your manuscript has been evaluated by the PLOS Biology editors, the Academic Editor and the original reviewers.

IMPORTANT: In addition, as previously advised, after the comments from the three original reviewers revealed a somewhat divergent assessment of the merits of your study, the Academic Editor asked us to secure advice from an Arbitrating Reviewer. This reviewer (reviewer #4), while being overall positive about your study, also recognises the need to address ore extensively the outstanding concerns of the other reviewers. Please accept our apologies for the additional time that this step has taken, but we do feel that it has been very helpful.

IMPORTANT: The Academic Editor has asked me to emphasise the need to seriously address the overlapping concerns raised by reviewers #1 and #2, in addition to the other requests from all the reviewers (including from reviewer #4). In addition, in order to help you understand the basis for our decision, I have included reviewer #4's arbitrating comments at the foot of the email. Specifically, the Academic Editor said "we should give authors the opportunity to revise, but I would emphasize that Reviewer#1 is very demanding and requested further analysis and that they need to do their best to respond and re-analyze data following the comments by reviewer#2 also. Somehow reviewer #1 and #2 share some thoughts. And of course, the issue raised by reviewer #4 on the neuronal adaptation is very relevant and at the very least needs to be considered."

In light of the reviews (below), we will not be able to accept the current version of the manuscript, but we would welcome re-submission of a much-revised version that takes into account the reviewers' comments. We cannot make any decision about publication until we have seen the revised manuscript and your response to the reviewers' comments. Your revised manuscript is also likely to be sent for further evaluation by the reviewers.

We expect to receive your revised manuscript within 3 months. 

**IMPORTANT - SUBMITTING YOUR REVISION**

*Re-submission Checklist*

*Published Peer Review*

*PLOS Data Policy*

*Blot and Gel Data Policy*

Sincerely,

Roli Roberts

Senior Editor,

rroberts@plos.org,

PLOS Biology

REVIEWERS' COMMENTS:

Reviewer #1:

The authors chose to rebuttal the arguments of three of the four reviewers but unfortunately did not provide the requested additional analyses that could have helped to support the authors' arguments. I appreciate the authors arguments. However, I would like to reemphasize the main problem that reviewer 2 and myself identified: Based on the currently presented data, the authors cannot state that there is a superior processing efficiency for non-alarm vs. alarm screams in general. This efficiency can only be stated for the task under investigation. Other tasks on the identical stimuli may reveal other results. E.g., the authors now provide the RT for alarm ratings that unsurprisingly do not show a superior processing efficiency for non-alarm calls when judging the alarm quality. This information should be included and discussed in the manuscript. Which unit is on the y-axis of the illustration in the response to reviewers; is it seconds? I strongly recommend discussing the significance of the current findings and relate these findings to previous reports in the literature showing, for example, a higher processing efficiency in terms of RT for alarm compared to non-alarm signals when judging the alarming quality. 

Another major point that still lacks attention is that the selection procedure has not been sufficiently justified. The authors argue that the selection was performed such that the recognition rates did not differ significantly. However, in the other sample of 33 participants, the authors elaborate on exactly those differences. I am afraid I do not understand the logic behind this approach. It would help if the authors clarified this point and show that the results in experiment 2 are robust against replacement of the selected stimuli by other stimuli which also conform to the selection criteria. The reference to Figure S1 is missing from the appropriate text passage, the stats for the n=26 group analyses similarly.

In the same line, the authors presented a good rationale for the standardization procedure but did not study whether this procedure affected the alarm vs. non-alarm screams and the scream types differentially. Even if it did, I believe that there could be something to learn about scream processing. If adjusting loudness affects alarm vs. non-alarm scream processing differently, this would identify loudness as a relevant (co-)factor in scream recognition.

The authors now state on page 8 that there was no significant difference between non-alarm and alarm screams in accuracy, but on page 9, following the subtitle "Impaired perceptual discrimination of alarm compared to non-alarm screams" they claim to have found such a difference. I believe the authors refer to the F-Test that includes the "neutral" screams. However, put in this context and given the post hoc results, this sentence is simply wrong.

One point I did not make sufficiently clear in my last review is that I believe that the observation of "two separate frequency bands" for affective screams simply results from the subtraction of MPS-values for the "neutral" screams. It is likely that the difference to "neutral" screams is rather a broader modulation for affective compared to neutral screams.

The authors did not reply to the question why loudness was included as a feature in the SVM when this parameter was actually standardized. 

The error rate in the gender identification task has not been sufficiently explored. Although the category factor did not reach significance (p=.132) the highly significant effect of scream type was not followed up. It suggests an unexpected interaction between scream type and task performance in case scream type identification and gender recognition were really orthogonal tasks.

On page 16 the authors overinterpret their findings as "specific neural pathways" and "specific decoding" without having tested specificity.

The Figure Legends should include the number of observations (n=26, 33…).

Reviewer #2:

I thank the authors for the careful revision and their responses to my comments. They raise some fair arguments and I agree with most. Yet, I still disagree with the strong claim that they are making that alarm screams have lower efficiency in the cognitive, neural, and communicative processing. It seems clear to me, that, based on the results presented by the authors, non-alarm screams have a higher discriminability advantage compared to alarm screams among themselves (among these three categories). This however, does not mean that alarm screams have a disadvantage. As reviewer 4 and also the authors themselves point out, 'alarm scream categories have some primacy during misclassification of other scream types' , which seems contradictory with the claim that the authors are trying to make. 

The authors argue in their response to one of my comment that 'discrimination within alarm screams can be lifesaving (e.g. mistaken an aggressive anger scream for a pain scream might cause you big harm, because you then tend to approach rather than to avoid the angry person/source).'; and 'organisms needs to decide quickly if a stream results from pain (help the other person; you do not need to run away from a person screaming out of pain) or from anger (run away from the angry person).' This differential behavioural response to sounds of pain and anger (approach someone in pain and run away from an angry person) would likely not occur in most non-human animals, or even in humans exposed to a dangerous environment (except if the person in pain is a close relative, which was not the case in your experiment). In nature, a pain scream might indicate that there is a high danger around (e.g. predator) and that one needs to run away (not necessarily approach the person/animal in pain), at least if we consider how non-human animals might react. Therefore, I still believe that in a natural setting (and likely in our ancestors), the immediate, safest response to any alarming sound would be to run away, before 'higher' cognitive processes occur for a finer discrimination.

The authors also argue that 'a stress response might explain increased RTs for alarm screams, but it does not explain higher error rates for alarm screams; if people take longer to classify alarm screams ("higher cognitive processing"), it does not explain why they make more errors (if people take longer to process sensory information, the usually get better in classifying the stimulus).' However, if I understand correctly, the higher errors (false alarms) were among alarm screams, showing that people find it hard to discriminate among these screams. Yet, they might still be able to quickly discriminate between alarm and non-alarm, which is what would be important for survival. Therefore, I think the following claim that the authors added should be revised: 

'While this could explain the increased classification times for alarm screams (i.e. respond first, and then discriminate), it does not explain the higher error rates, which still point to a classification and discrimination disadvantage for alarm screams.'

I appreciate that the authors now also quantified the RTs to the alarm ratings presented in Fig. 1c, and found that people did not show faster RTs in the alarm rating for alarm compared to non-alarm or neutral screams. However, these new results also show that people were not slower for rating the alarm of alarm screams, suggesting no 'inferior processing efficiency' for these types of sounds.

Overall, I still think that these results warrant publication, but I think the authors should be more cautious with how they interpret their results. For instance, I would tone down claims suggesting that alarm scream have some disadvantages, and develop potential alternative hypotheses further in the discussion. They could make clear, throughout the manuscript, that the impaired perceptual discrimination of alarm screams is among the three types of screams of this category (not between alarm and non-alarm screams). In addition, it would be clearer if the term 'reaction time' could be changed to 'response time' or 'decision time', since this measure does not assess how fast people 'reacted' (they might have reacted before choosing an answer) but how fast people responded.

Reviewer #3: 

[identifies himself as Harold Gouzoules]

I believe Frühholz et al. have improved the manuscript considerably and they have explicitly acknowledged virtually all of the concerns the reviewers outlined, myself included. In most instances, I think the authors have provided very thoughtful responses, but I will not weigh in here as to whether I think the issues raised by the other reviewers have been adequately addressed in this revision. Instead, I'll focus on the central issues I raised in my initial review that, I respectfully suggest, still need some additional attention.

My main concern remains with the source of the screams used in the experiment. My initial review noted the authors' description:

"We thus instructed the 12 participants to produce vocalizations of screams for each of the 6 types of generic screams, as well as to produce neutral screams on the basis of an intense vocalization of the vowel /a/. For each type of scream, we provided short instructions to each participant to imagine several corresponding context(s) in which these screams are commonly produced (e.g. fearful scream: "You are being attacked by an armed stranger in a dark alley"; anger screams: "You try to intimidate an opponent"; joyful screams: "Your favorite team wins the World Cup"; pleasure screams: "You are screaming from sexual delight"). For each participant, we recorded 8 instances of each type of scream in an anechoic chamber with a Neumann TLM-102 microphone at a distance of ~1m from the speaker. From these 8 instances, we chose the 5 best recordings from perceptual judgment of recording quality …"

I think that the full set of prompts ("short instructions") given to the participants should be provided in the paper or supplemental materials. I assume those listed in the manuscript are just examples. What were the others for the different scream types? With a prompt such as "Your favorite team wins the World Cup" --- were the instructions to produce a scream the participant might imagine a rabid fan would make, or simply to produce a scream that they themselves would make? If the latter, what if the person is not a football fan, or even a sports fan? That would surely generate variation in the screams produced in this context that those involving, for example, the intense fear context might not show. Similarly, for the prompt "You are screaming from sexual delight," is the assumption that males and females are equally likely to scream in this context, or that, within one sex there are no differences in the tendency to scream during sex or in the nature of any screaming? How might the study's results be impacted by these issues?

In my initial review I noted:

"The study's stimuli come from 12 volunteers asked to generate screams that they think would be associated with six different scenarios. Nothing is said about these participants (other their mean ages and that that they were healthy. Problematically, the authors assume that these 12 individuals have the acting talent to generate different screams as they were asked to do. …. Engelberg and Gouzoules note: "It is important to emphasise that the chance-level performances demonstrated here do not necessarily imply that acted and natural screams are identical in every respect, nor that screams from all actors are equally suitable for implementation in empirical research."

To this, Frühholz et al. respond:

"If screams are within the communication repertoire of every human, it should be that not only talented actors (NB: being an actor on TV might not necessarily imply that someone is talented) are able to produce valid vocalizations of screams, but every normal human being with precise instructions. We took care that scream vocalizations were produced at the intended level of screaming."

I have no idea what the authors mean when they say, "We took care that scream vocalizations were produced at the intended level of screaming." This is a vague statement. Could they please elaborate? The contention that "If screams are within the communication repertoire of every human, it should be that not only talented actors (NB: being an actor on TV might not necessarily imply that someone is talented) are able to produce valid vocalizations of screams, but every normal human being with precise instructions" is simply not convincing to me. The authors' rib that not all TV actors are talented misses the point I was trying to make. We might indeed debate the talent of any individual TV actor, but there is no denying that these individuals are professionals under the guidance of a director and they have the opportunity more rehearsals and multiple takes. The authors are clearly reluctant to accept my point that it is an unfounded, and I suggest unreasonable, assumption that every human being is equally capable of producing different sorts of screams (or other nonverbal expression) that reflect natural usage, upon command …. no more than, despite having the general capacity for language, all humans can convincingly deliver a Shakespearean soliloquy. That screams are in the human vocal repertoire does not mean that they can be generated on command without training and experience …. that's the point and caveat we make (in Engelberg & Gouzoules 2018) about the potential to produce credible screams by professional actors.

Again, as I noted in my earlier review, the authors provide no details about how the screaming participants in this study were recruited? I think it's critical to the interpretation of the results to have more details. Did any have acting experience, for example? Are the results of the experiments comparable when examined as a function of the screams clusters contributed by the different screamers (i.e., were results associated with any of the scream contributors at odds with the general findings reported)? 

In the response to this issue the authors go on to say:

"Furthermore, give the acoustic similarities our screams to screams reported in other papers, we are confident to have a recorded a scream database with our speakers that is of a valid nature."

This is also vague. To my knowledge, other papers have not explored in detail screams associated with the variety of contexts examined in the Frühholz et al. study, so how can the authors make this claim? Please elaborate with specific details.

The authors continue their response with:

"Within the field of voice signaling research there is a big discussion if studies should use more natural vocalizations (with the disadvantage of having sampling biases, noisy backgrounds, unequal quality between signal categories, etc.) or using acted and recorded voice signals in the lab. Only the latter allows a more precise sampling of vocalizations and database creations, precise vocalization instructions, clean recordings, and equal sampling of vocalizations across categories. Since in this study we were interested in conducting precise psychoacoustic experiments and also precise neuroimaging experiments (which can be easily affect by degraded stimulus quality), we opted for using acted scream vocalizations, given that there are perceptually very close to natural screams (see quote above)."

I am, of course, aware of this issue in the literature (we review it briefly in Engelberg & Gouzoules 2018), and I appreciate that it has major implications for study design, and also for much significant theory that underpins ideas about human communication. However, the choice to use acted renditions of screams generated by (presumably) untrained participants is nonetheless an issue that should not remain unaddressed in the manuscript. I would ask that the authors, at the very least, acknowledge the possibility that an inability (by all, or some participants, or males, or female …) to produce naturalistic screams for the contexts described in the prompts might have impacted the results. To some degree, this question can be explored in the data, and I encourage the authors to examine this in whatever ways possible. 

I have just a few other comments, in particular, about the summary of the comparative literature that the authors have now included in the manuscript. They say:

"Screams by lower ranking animals help to recruit support from allies [14,15], while higher ranking animals scream to intimidate the lower ranking [16]." 

This is actually more complicated. Screams are not always used by dominants in agonistic encounters: there are separate and distinct threat vocalizations (at least in the various macaque and baboon species) that are acoustically very different from screams. But when challenged by a lower-ranking opponent, high-ranking individuals will revert to specific screams (we designated them as "arched screams") that are used to recruit support from matrilineal family members.

Finally, in response to the additional literature suggestions I made, the authors note that " …. this list seems limited towards the reviewers' own work." I apologize if my literature suggestions came across as trying to promote my research. I think it's fair to say that ours is the primary extended body of work on animal screams. A key takeaway is that that "alarm" is not the only thing communicated by nonhuman primate screams (see above observations about the use of screams by domain monkeys). Even in nonhuman primates, screams are more complicated acoustically, and in their contextual usage, than in most other taxa (acknowledging, of course, the authors' that the broad context of monkey and ape screams, unlike for humans, is agonistic in nature … a point I have been making in print and publicly, in interviews available on the web, for years).

Harold Gouzoules

Reviewer #4:

Fruhholz et al present a comprehensive investigation on screams. In particular, they extend the investigation to non-alarming screams and present a series of behavioral and neuroimaging studies. They observe that alarming scream are processed less efficiently and lead to smaller BOLD responses. Overall this is an impressive study, nicely written and very comprehensive. The results run contrary to expectations, which make the study the more interesting and potentially impactful. Because results are somehow unexpected, caution on the interpretation and duly ruling out alternatives is particularly important. Along those lines, I would like to understand better the fMRI study and in particular the effect of the null event (baseline) as well as neural adaptation. Below I provide a further description of those issues and other that the authors might consider.

I was very surprised to see in the fMRI results such a strong activation for the neutral stimuli. Can the authors comment in this surprising result and on how that speaks for the efficiency of the fMRI design.

One alternative for the low responses observed for alarming sounds is neuronal adaptation. A known physiological fact is that stronger initial responses lead to smaller subsequent responses. Since neural adaptation can explain in part the surprising low responses reported in the fMRI, I invite the authors to provide an analysis taking into account the serial order of the conditions. Here, one can compare the amplitude of the response for each category for the first, second, third, etc presentation. 

Experiment 2 and 3. Please provide the intrasubject consistency (reliability when responding twice to the same stimuli), as well as intersubject agreement per stimuli and categories.

Experiment 4: Provide statistic for the reliability of the results comparing the post experimental test in experiment 4 with those obtained in the other studies. How replicable are the behavioral results across the two populations of subjects?

Methods:

Experiment 1: describe the subject recruited: where those naïve subject, or professional actors?

As there is no apriori reason to think that the authors have exhaustively cover all possible screams, I suggest to modify the following sentence " to comprehensively cover all possible screams…" to some like "to cover a broader range of possible screams". 

Page 24. Using how quality headphones modify for using high quality…

Provide descriptive statistics for missing trials, where those equally distributed across all the studied categories?

RT - outlier removal. What there any procedure to remove outliers?

For the calculation of d' explain what was taken as the noise and the signal distribution for each contrast. 

Specify what is the implicit baseline for the fMRI study. For the methods, it wasn't clear whether a null event was incorporated in the fMRI design. provide length of each blocks

Figures:

Fig 2. B and C labels are mixed up

Results:

F statistics: some of the reports of degrees of freedom do not check with the reported sample size. Please revise.

ARBITRATOR COMMENTS FROM REVIEWER #4 [lightly edited]:

Now I had the chance to review the paper. It is indeed a very comprehensive one and it is clear the authors have put a lot of work into it i.e., constructing new stimulus materials, behavioral studies and also fMRI.

however after reading the manuscript, the reviewers comments and the replies I must admit that I am on the reviewers' side.

Overall, I believe the points all 3 reviewers are making are reasonable and fair. They do raise serious concerns that muddle the interpretation of the data; and in my view the authors could minimally have toned down the interpretation and/or best run a control study.

Moreover, the paper is quite vague, or silent on critical aspects of the studies, i.e., the methods. For replicability, specially of this new and unexpected findings, the authors should make an effort to better and more extensively describe the experimental procedure.

The reviewers have raised some of these points, but the reply was not satisfactory. For instance,

1) stimulus materials: what are the specific instructions given to the participant to prompt the stimuli? 

Who where those subjects: professional actors or naive participants. 

2) fMRI design: I couldn’t tell whether they have a null event or not, and/or what was the baseline. It is rather surprising that the highest activations are for the neutral stimuli, this makes me to suspect strong neural adaptation. If so, then an important part of the result might just be confounded by the design

3) standarization: Reviewer 1 makes a very good point that the standardisation could explain part of the results. The authors have decided to argue but this is a serious point as all results could boil down to alarm scream being higher in loudness and the brain normalising for this. If this feature is removed from the stimuli then they loose the critical property. The authors have remove that for good reasons, as it may confound the fMRI result. However, a better alternative could have been to include it in the GLM and investigate it explicitly.

4) stimuli subselection: from the 420 stimuli that were initially created a 1/4 is used for the further studies but the rationale for selecting those is poorly described. Furthermore, given the surprising results, demonstrating that the behavioral results are not driven by the subselection would have been a much better choice.

[in the reviewer comments], I am providing you with further aspects for the authors to consider beyond the ones that have been already raised.

Overall, this is a nice manuscript; and the results (if true), are interesting. If the authors could provide a more compelling revision (addressing the concerns raised by the reviewers for instance by explicitly acknowledging the effect of the task), it could be nice paper for PloS Biology.

---

## [Decision Letter · Decision Letter 2]

29 Jan 2021

Dear Dr Frühholz,

Thank you for submitting your revised Short Report entitled "Neurocognitive processing efficiency for non-alarm rather than alarm signaling in human scream calls" for publication in PLOS Biology. I have now obtained advice from three of the original reviewers and have discussed their comments with the Academic Editor. 

Based on the reviews, we will probably accept this manuscript for publication, assuming that you will modify the manuscript to address the remaining points raised by the reviewers. Please also make sure to address the data and other policy-related requests noted at the end of this email.

IMPORTANT:

a) Please address the remaining requests from reviewers #1 and #4.

b) This paper has been considered as a Short Report, for which the maximum number of Figures is 4. However, we note that you now have 7. Please could you combine several of the Figures, as you feel appropriate (I note that Figs 2, 3, 4, 5 and 7 are small, with a relatively simple structure, so could easily be incorporated into adjacent Figs).

c) Please could you improve the title? We suggest "Neurocognitive processing efficiency for discriminating between human non-alarm and alarm scream calls" or "Human neurocognitive processing efficiency for discriminating between non-alarm and alarm scream calls" - please note reviewer #1's comment regarding the title.

d) Regarding your Data Availability Statement, please could you revert to wording more akin to the previous version, which explained the reasons for the restricted data access [ “Due to Swiss legal and ethical restrictions at the time of data sampling for this study (i.e. full consent of participants required for public sharing), data can only be made available from the corresponding author upon reasonable requests.”]. Please also note that we discourage the use of a single named contact, as this lacks stability in perpetuity; please see PLOS guidelines here: https://journals.plos.org/plosbiology/s/data-availability#loc-acceptable-data-sharing-methods

e) Please address my further Data Policy requests further down, regarding the data directly underlying the Figures.

We expect to receive your revised manuscript within two weeks. Your revisions should address the specific points made by each reviewer. 

-  a cover letter that should detail your responses to any editorial requests, if applicable

*Published Peer Review History*

*Early Version*

Sincerely,

Roli Roberts

Senior Editor,

rroberts@plos.org,

PLOS Biology

ETHICS STATEMENT:

-- Please include information about the form of consent (written/oral) given for research involving human participants. All research involving human participants must have been approved by the authors' Institutional Review Board (IRB) or an equivalent committee, and all clinical investigation must have been conducted according to the principles expressed in the Declaration of Helsinki.

DATA POLICY:

Regardless of the method selected, please ensure that you provide the individual numerical values that underlie the summary data displayed in the following figure panels as they are essential for readers to assess your analysis and to reproduce it: Figs 1ABCDE, 2AB, 3ABCDEFGH, 4ABCD, 5AB, 6ABCDEF, 7, S1ABC, S2, S3. I see that you're provided a zipped folder for data, but this only seems to contain Microsoft Access shortcuts, a few kb in size. NOTE: the numerical data provided should include all replicates AND the way in which the plotted mean and errors were derived (it should not present only the mean/average values).

REVIEWERS' COMMENTS:

(these should be the correct numbering, as per the original version; apologies for the previous reviewer numbering error)

Reviewer #1:

I feel that the authors have made a nice job in answering the reviewers' suggestions and I think them for their effort. 

I am satisfied with the results from the replication experiment that suggests that stimulus selection does not represent a relevant bias. Also the non-significant effect of original loudness of stimuli suggests that the observed effects are not primarily driven by the standardization procedure. 

The current manuscript is much more balanced in the sense that the authors acknowledge the importance of the task under investigation (which may also party explain the relative strong fMRI response to neutral stimuli during the gender discrimination task - an interpretation that should be added to the manuscript).

The only two parts of the manuscript that still allow for misinterpretation are the title and the abstract. I believe that "Neurocognitive processing efficiency for discriminating human non-alarm rather than alarm scream calls" would not diminish the significance of the paper in any way and more specifically describe what the paper is mainly about. In the same line, the last sentence of the abstract should highlight what the authors can indeed claim, for example: ...in humans and that any potential threat processing bias does not translate into a higher efficiency during scream discrimination or into increased implicit processing when identifying the screaming person's gender."

Reviewer #2:

The authors present a good revision of their manuscript. They are now much more cautious with how they interpret their results, and integrated some alternative explanations for these results in the discussion. I am happy with how they dealt with my comments and have no further remarks. 

Reviewer #4:

[identifies himself as Harold Gouzoules]

Frühholz et al. have made brief changes in this revision that are consistent with, and largely responsive to, my comments on the previous draft. Ideally, additional fleshing out of some of the points would have been desirable, but because the article is to be of the "short report" form, I won't ask for additional changes other than a couple of points noted below.

The authors have very nicely responded to my request for a more accurate account of the nonhuman primate literature on the use of screams during agonistic encounters. One very slight change would enhance the account. In the revision, they say:

"Screams by lower-ranking animals help to recruit support from allies [18,19], while higher-ranking animals scream to intimidate the lower-ranking animal when challenged by this lower-ranking opponent [20]".

The use of screams by dominant monkeys when challenged by a lower-ranking individual is not a matter of intimidation. Instead, those screams also recruit support from matrilineal kin … just as screams do for subordinate monkeys attacked by a dominant individual. The difference is that a monkey will use an acoustically different scream when confronted by a dominant opponent compared to when it is challenged by a subordinate individual. The different types of screams thus provide listeners with information about the nature of the fight, and the kind of response (intervention and assistance) they show is influenced by the scream type they hear.

Although I have not previously "weighed in" on the other reviewers' comments or Frühholz et al.'s responses to them, it seems to me that, in the case below, the matter debated could be resolved by the proposal that the recognition accuracy question makes more sense from an evolutionary cost-benefit perspective. The issue:

Reviewer 2 says: " ….. It seems clear to me, that, based on the results presented by the authors, non-alarm screams have a higher discriminability advantage compared to alarm screams among themselves (among these three categories). This however, does not mean that alarm screams have a disadvantage. …."

Frühholz et al. respond: "Furthermore, the observation that alarm screams categories are often used during misclassification of all screams ('alarm scream categories have some primacy during misclassification of other scream types') can be a topic of debate. If alarm scream categories are often used during misclassification of any type of scream, this again does not speak for a processing "advantage". A cognitive system usually wants to achieve high recognition accuracy, so introducing a factor in the recognition process that facilitates classification errors is surely not of "advantage" for a human recognition system. So, we are unsure what the reviewer means by stating that this 'seems contradictory with the claim that the authors are trying to make'."

I suggest that "errors" or perceptual bias in a communication system might be advantageous from an evolutionary perspective, one that goes beyond a view rooted exclusively in cognitive efficiency and accuracy. Specifically, given the evolutionary origins of screams (calls given by prey in the grasp of a predator) and the associated emotional state of fear, processing errors that bias "non-alarm screams" toward "alarm screams" likely have less cost than the reverse error (i.e., failing to process a true "alarm scream" as such). The adaptive response or "bias" is the one where the errors have less cost, and misclassifying a non-alarm scream as an alarming one is likely less harmful than perceiving an alarm scream as non-alarm. I think the "advantage - disadvantage" disagreement above is thus misguided. Frühholz et al.'s persistence with the interpretation that alarm screams are disadvantaged is the main source of confusion here, I think, and along with reviewer 2, I consider the conclusion problematic.

---

## [Editor Report · Decision Letter 3]

15 Feb 2021

Dear Dr Frühholz,

On behalf of my colleagues and the Academic Editor, Manuel Malmierca, I'm pleased to say that we can in principle offer to publish your Short Report "Neurocognitive processing efficiency for discriminating human non-alarm rather than alarm scream calls" in PLOS Biology, provided you address any remaining formatting and reporting issues. These will be detailed in an email that will follow this letter and that you will usually receive within 2-3 business days, during which time no action is required from you. Please note that we will not be able to formally accept your manuscript and schedule it for publication until you have made the required changes.

PRESS: We frequently collaborate with press offices. If your institution or institutions have a press office, please notify them about your upcoming paper at this point, to enable them to help maximise its impact. If the press office is planning to promote your findings, we would be grateful if they could coordinate with biologypress@plos.org. If you have not yet opted out of the early version process, we ask that you notify us immediately of any press plans so that we may do so on your behalf.

Thank you again for supporting Open Access publishing. We look forward to publishing your paper in PLOS Biology. 

Sincerely,

Roli Roberts

Roland G Roberts, PhD 

Senior Editor 

PLOS Biology